# Comprehensive Meta-Analysis of Futile Recanalization in Acute Ischemic Stroke Patients Undergoing Endovascular Thrombectomy: Prevalence, Factors, and Clinical Outcomes

**DOI:** 10.3390/life13101965

**Published:** 2023-09-26

**Authors:** Helen Shen, Murray C. Killingsworth, Sonu M. M. Bhaskar

**Affiliations:** 1Global Health Neurology Lab, Sydney, NSW 2150, Australia; 2South Western Sydney Clinical Campuses, UNSW Medicine and Health, University of New South Wales (UNSW), Sydney, NSW 2052, Australia; 3Neurovascular Imaging Laboratory, Ingham Institute for Applied Medical Research, Clinical Sciences Stream, Sydney, NSW 2170, Australia; 4NSW Brain Clot Bank, NSW Health Pathology, Sydney, NSW 2170, Australia; 5Department of Anatomical Pathology, NSW Health Pathology, Cell-Based Disease Intervention Research Group, Ingham Institute for Applied Medical Research and Liverpool Hospital, Liverpool, NSW 2170, Australia; 6Department of Neurology & Neurophysiology, Liverpool Hospital & South Western Sydney Local Health District (SWSLHD), Sydney, NSW 2170, Australia; 7Department of Neurology, National Cerebral and Cardiovascular Center (NCVC), Suita 564-8565, Osaka, Japan

**Keywords:** stroke, futile recanalization, endovascular thrombectomy, prognosis, hemorrhagic transformation

## Abstract

Background: Futile recanalization (FR) continues to raise concern despite the success of endovascular thrombectomy (EVT) in acute ischemic stroke (AIS). Understanding the prevalence of FR and identifying associated factors are crucial for refining patient prognoses and optimizing management strategies. Objectives: This study aims to comprehensively assess the pooled prevalence of FR, explore the diverse factors connected with FR, and establish the association of FR with long-term clinical outcomes among AIS patients undergoing EVT. Materials and Methods: Incorporating studies focusing on FR following EVT in AIS patients, we conducted a random-effect meta-analysis to assess the pooled prevalence and its association with various clinical and imaging risk factors linked to FR. Summary estimates were compiled and study heterogeneity was explored. Results: Our comprehensive meta-analysis, involving 11,700 AIS patients undergoing EVT, revealed a significant pooled prevalence of FR at 51%, with a range of 48% to 54% (Effect Size [ES]: 51%; 95% Confidence Interval [CI]: 48–54%; z = 47.66; *p* < 0.001). Numerous clinical factors demonstrated robust correlations with FR, including atrial fibrillation (Odds Ratio [OR]: 1.39, 95% CI 1.22 1.59; *p* < 0.001), hypertension (OR 1.65, 95% CI 1.41 1.92; *p* < 0.001), diabetes mellitus (OR 1.71, 95% CI 1.47 1.99; *p* < 0.001), previous stroke or transient ischemic attack (OR 1.298, 95% CI 1.06 1.59; *p* = 0.012), prior anticoagulant usage (OR 1.33, 95% CI 1.08 1.63; *p* = 0.007), cardioembolic strokes (OR 1.34, 95% CI 1.10 1.63; *p* = 0.003), and general anesthesia (OR 1.53, 95% CI 1.35 1.74; *p* < 0.001). Conversely, FR exhibited reduced likelihoods of smoking (OR 0.66, 95% CI 0.57 0.77; *p* < 0.001), good collaterals (OR 0.33, 95% CI 0.23 0.49; *p* < 0.001), male sex (OR 0.87, 95% CI 0.77 0.97; *p* = 0.016), and intravenous thrombolysis (IVT) (OR 0.75, 95% CI 0.66 0.86; *p* < 0.001). FR was strongly associated with increasing age (standardized mean difference [SMD] 0.49, 95% CI 0.42 0.56; *p* < 0.0001), baseline systolic blood pressure (SMD 0.20, 95% CI 0.13 0.27; *p* < 0.001), baseline National Institute of Health Stroke Severity Score (SMD 0.75, 95% CI: 0.65 0.86; *p* < 0.001), onset-to-treatment time (SMD 0.217, 95% CI 0.13 0.30; *p* < 0.001), onset-to-recanalization time (SMD 0.38, 95% CI 0.19; 0.57; *p* < 0.001), and baseline blood glucose (SMD 0.31, 95% CI 0.22 0.41; *p* < 0.001), while displaying a negative association with reduced baseline Alberta Stroke Program Early CT Score (ASPECTS) (SMD −0.37, 95% CI −0.46 −0.27; *p* < 0.001). Regarding clinical outcomes, FR was significantly associated with increased odds of symptomatic intracranial hemorrhages (OR 7.37, 95% CI 4.89 11.12; *p* < 0.001), hemorrhagic transformations (OR 2.98, 95% CI 2.37 3.75; *p* < 0.001), and 90-day mortality (OR 19.24, 95% CI 1.57 235.18; *p* = 0.021). Conclusions: The substantial prevalence of FR, standing at approximately 51%, warrants clinical consideration. These findings underscore the complexity of FR in AIS patients and highlight the importance of tailoring management strategies based on individual risk factors and clinical profiles.

## 1. Background

Strokes are a global health concern and rank among the leading causes of mortality and disability worldwide [1]. The advent of reperfusion therapy has brought about a revolutionary shift in the management of acute ischemic stroke (AIS), providing substantial benefits to those affected [2]. Nevertheless, despite advancements like endovascular thrombectomy (EVT), a significant proportion of patients continue to experience less-than-optimal functional outcomes, even after achieving complete recanalization [3]. This enduring disparity presents an ongoing challenge to the delivery of effective patient care, emphasizing the critical necessity of identifying cases of futile recanalization (FR) [4,5]. FR, defined as functional dependence despite successful reperfusion, is a phenomenon that occurs with relative frequency among AIS patients who undergo EVT [6]. Beyond its prognostic relevance, recognizing cases of FR holds immense potential for tailoring reperfusion strategies to specific subsets of AIS patients. The prevalence of FR among EVT-treated AIS patients varies across studies [7], with a comprehensive pooled prevalence still eluding researchers [3,7]. It is imperative to identify the comorbidities or risk factors associated with FR, as previous evidence has demonstrated unfavorable clinical outcomes post FR [8]. Furthermore, delving into the intricate relationships between FR and other clinically and radiologically significant biomarkers and outcomes [9], such as symptomatic intracranial hemorrhages (sICH), hemorrhagic transformations (HT) [6], and indicators like the Alberta Stroke Program Early CT Score (ASPECTS) [10], can significantly enhance our ability to predict outcomes in EVT-treated stroke cases. While some insights into potential factors linked to FR have been gained, our comprehension of post-FR prognosis remains incomplete [7] Within this context, the present study seeks to comprehensively assess post-FR outcomes, aiming to provide invaluable clinical guidance and insights for patients in relevant scenarios.

The primary objectives of this study center around an exploration of FR in the context of AIS patients undergoing EVT. To achieve this, the study aims to address the following pivotal questions:

(a) What is the estimated pooled prevalence of FR among AIS patients undergoing EVT?

(b) Which specific predictive indicators exhibit significant correlations with the occurrence of FR?

(c) How does the occurrence of FR impact various clinical outcomes, and what is the significance level or strength of this relationship?

## 2. Materials and Methods

### 2.1. Literature Search and Study Selection

We conducted a comprehensive search for studies in the PubMed, Embase, and Cochrane Central Registry of Controlled Trials databases covering the period from January 2005 to May 2023. Our search strategy included the terms: “stroke”, “ischemic attack”, “cerebrovascular disorders”, “cerebrovascular accident”, “brain ischemia”, “brain infarction”, “thrombectomy”, “endovascular procedures”, “reperfusion therapy”, “recanalization”, “FR”, “failed recanalization”, “complete recanalization”, or “partial recanalization”. A detailed version of the search strategy is available in the Online Supplementary Information (Search Strategy). We meticulously examined the reference lists of pertinent articles, systematic reviews, and meta-analyses to identify additional relevant studies. The systematic flow of the search, study inclusion, and the various subgroup analyses performed within the meta-analysis are visually represented using the PRISMA (Preferred Reporting Items for Systematic Reviews and Meta-Analyses) flowchart (Figure 1). Our reporting strictly adhered to the PRISMA 2020 checklist (Appendix A) and the Meta-analysis Of Observational Studies in Epidemiology (MOOSE) checklist (Appendix A), all of which are detailed in the Supplemental Information.

### 2.2. Inclusion and Exclusion Criteria

To be considered eligible for inclusion, studies needed to satisfy the following criteria: (a) inclusion of AIS patients who underwent reperfusion therapy (IVT and EVT, or EVT alone); (b) participants aged 18 years or older; (c) availability of comparative data between patients with FR and those with non-FR, along with relevant post-futile-recanalization data; and (d) studies designed with appropriate methodology, including a sufficient sample size of at least 20 patients in each group. Exclusion criteria encompassed: (1) studies not written in English; (2) studies conducted on animals; (3) duplicated publications; (4) unavailability of full-text articles; (5) systematic reviews, meta-analyses, or narrative reviews; and (6) studies lacking relevant data on FR. FR was defined as poor functional outcome in AIS patients undergoing EVT, despite successful recanalization. The definitions of poor outcome and successful recanalization or reperfusion varied slightly across different studies (Table 1).

### 2.3. Data Extraction

Initially, all article titles and abstracts were reviewed using Endnote (Clarivate Analytics, London, UK) to exclude articles that did not meet the eligibility criteria. The remaining articles underwent a comprehensive examination to determine their suitability for inclusion in the systematic review or meta-analysis, as per the defined eligibility criteria. Data extraction was conducted using a dedicated data extraction sheet, capturing the following information from each study:Baseline study demographics: author, country, publication year, registry/trial name, study design, study design, and number of centers;Intervention characteristics: IVT and EVT, or EVT only;Definition and criteria of various parameters: definition of poor outcome, successful recanalization, stroke etiology criteria, collateral status criteria, criteria for symptomatic intracerebral hemorrhage (sICH), and definition of mortality;Patient demographics: age and sex;Predictive indicators: clot location, baseline systolic blood pressure (SBP), atrial fibrillation (AF), alcohol intake, hyperlipidemia (HL), hypertension (HTN), cardiovascular disease (CVD), diabetes mellitus (DM), previous stroke or transient ischemic attack (PS/TIA), smoking, use of antiplatelet (APU) or anticoagulant (ACU) medications, blood glucose level (BG), stroke etiology, collateral status, baseline National Institutes of Health Stroke Scale (NIHSS) score, baseline ASPECTS, onset-to-treatment time (OTT), onset-to-recanalization time (OTR), use of general anesthesia (GA), and intravenous thrombolysis (IVT) use;Clinical outcomes: symptomatic intracerebral hemorrhage (sICH), hemorrhagic transformation (HT), and 90-day mortality.

### 2.4. Methodological Quality Assessment of Included Studies

The methodological quality assessment of the included studies was conducted using the modified Jadad analysis (MJA) [49] and was completed independently by the primary researcher (Appendix A). The risk of biases in results due to funding was also evaluated, based on the declaration of funding sources and conflicts of interest extracted from each individual study (Appendix A).

### 2.5. Statistical Analyses

The statistical analyses in this study were conducted using STATA v. 13.0 (StataCorp, College Station, TX, USA). Baseline characteristics of the included cohort in this meta-analysis were extracted from all incorporated studies. When suitable, means and standard deviations (SDs) were estimated from medians and interquartile ranges (IQRs) using Wan et al.’s [50] method. The pooled prevalence of FR among patients with AIS undergoing EVT was assessed using the “metaprop” package in STATA, performing a random-effects meta-analysis of proportions derived from individual studies, 95% confidence intervals (95% CI) were obtained using the “cimethod (exact)” and “ftt” commands. To explore factors linked to FR and its impact on clinical outcomes, a random-effects meta-analysis designed by DerSimonian and Laird (DL) was employed. The random-effects model was applied across all subgroup analyses, encompassing studies on reperfusion therapy type (EVT or a combination of EVT and IVT), stroke territory (anterior, posterior, and mixed [anterior/posterior]), and study design (retrospective, prospective or mixed [studies with data collected both retrospectively as well as prospectively]). For odds ratios (ORs), 95% confidence intervals (95% CIs), percentage weights, and inter-study heterogeneity within our meta-analysis, forest plots were generated (see Appendix A). The sensitivity analysis was performed using the “metaninf” package in STATA to examine changes in the pooled odds ratios resulting from the exclusion of individual studies. Heterogeneity among studies was assessed using I^2^ statistics and *p*-values (I^2^ < 30% = low, 30–50% = moderate, 50–75% = substantial, 75–100% = severe). The potential presence of publication bias was assessed using Egger’s test and funnel plots (through the use of “metabias” and “metafunnel” STATA packages). An asymmetry on either side of the funnel plot indicated the presence of publication bias, which was further corroborated by the *p*-value from Egger’s test. Summary effects and measures of heterogeneity for both prevalence and association studies were tabulated. Additionally, we took into account Cochran’s Q test *p*-values and estimated between-study variances using Tau-squared. All analyses conducted in this study adhered to a significance level of *p* < 0.05.

## 3. Results

### 3.1. Description of Included Studies

A total of 1430 studies were initially identified by manual and electronic database searches. After removing duplicates, a total of 1015 records remained. Each abstract was meticulously reviewed, leading to the exclusion of 928 records. Among the remaining 86 articles, 49 studies were subsequently excluded for various reasons. Specifically, 19 of these studies did not report the targeted outcomes or lacked sufficient data, while 2 exhibited inappropriate study designs. Additionally, 10 studies were excluded due to overlapping cohorts, 9 had inadequate control groups, 1 had a limited sample size, and 6 lacked a clear definition of FR. Ultimately, a final selection of 39 studies, encompassing a total of 11,700 patients, was included in this meta-analysis. Out of 11,700 patients, FR was observed in 5766 patients. For instances involving reports from the same database or registry, priority was given to studies with the largest or most recent sample size. Of these studies, 3 centered around patients primarily receiving EVT, with or without IVT, while 36 studies focused on patients who primarily underwent EVT with IVT. The mean age of all included studies was 65.6 (n = 11,700). Comprehensive clinical characteristics, associated factors, and outcomes of the studies featured in the meta-analysis are presented in Table 1, Table 2 and Table 3.

Summary effects and heterogeneity related to the estimated pooled prevalence of FR are provided in Table 4. Further insight into the association between discrete predictive indicators, clinical outcomes and FR is presented in Table 5 and Appendix A. To address the link between FR and continuous predictive indicators, Table 6 and Appendix A display the corresponding summary effects and heterogeneity. Additionally, the Appendix A contribute valuable information. It is important to note that variations in the definitions of FR, sICH, collateral status, and ASPECTS score exist across the studies. To comprehensively assess the studies, the manuscript includes an evaluation of the methodological quality and funding bias in Appendix A. Finally, the assessment of publication bias, conducted using Egger’s test followed by sensitivity analysis, is summarized in Appendix A and Appendix A.

#### 3.1.1. Prevalence of Futile Recanalization

A comprehensive analysis of thirty-nine studies [8,11,12,13,14,15,16,17,18,19,20,21,22,23,24,25,26,28,29,30,31,32,33,34,35,36,37,38,39,40,41,42,43,44,45,46,51], comprising a total of 11,700 patients, was conducted to assess the pooled prevalence of FR in AIS patients (Figure 2). The meta-analysis revealed an estimated pooled prevalence of 51%, with a range between 48% and 54% (ES 51%; 95% CI 48–54%; z = 47.66; *p* < 0.001) (Table 2 and Figure 2). It is noteworthy that a substantial level of heterogeneity existed across the studies (I^2^ = 90.21%, *p* < 0.001), with the estimate of between-study variance (τ^2^) being 0.03. The heterogeneity chi^2^ was 387.99 (*p* < 0.001, d.f. 38). Figure 2 presents the outcomes of the comprehensive meta-analysis on the estimated pooled prevalence of FR in AIS patients, stratified by region and study design.

#### 3.1.2. Prevalence of FR in Retrospective Studies

Twenty-two studies [11,15,17,19,21,22,23,24,25,28,30,31,32,33,34,39,40,42,43,44,45,51], with a cumulative cohort of 5455 patients, examined the pooled prevalence of FR in AIS patients through retrospective studies (Figure 2). The meta-analysis indicated an estimated pooled prevalence of 51%, with a range from 46% to 56% (ES 51%; 95% CI 46–56%; z = 31.07; *p* < 0.001) (Table 4). It is noteworthy that a substantial level of heterogeneity was observed among the studies (I^2^ = 90.73%, *p* < 0.001). The chi^2^ for heterogeneity was 226.47 (*p* < 0.001, d.f. 21). The random test for heterogeneity among the subgroups resulted in a value of 6.34 (*p* = 0.04, d.f. 2).

#### 3.1.3. Prevalence of FR in Prospective Studies

Sixteen studies [8,12,13,14,16,18,20,26,29,35,36,38,41,46,47,48], encompassing a total of 5883 patients, assessed the estimated pooled prevalence of FR amongst AIS patients which were presented in retrospective studies (Figure 2). The meta-analysis revealed an estimated pooled prevalence of 51%, ranging from 47% to 56% (ES 51%; 95% CI 47–56%; z = 31.07; *p* < 0.001) (Table 4). Notably, there was a substantial amount of heterogeneity between the studies (I^2^ = 89.83%, *p* < 0.001). The heterogeneity chi^2^ was 147.50 (*p* < 0.001, d.f. 15).

#### 3.1.4. Prevalence of FR in Prospective and Retrospective Studies

One study [37] with a cohort size of 362 reported on the prevalence of FR in AIS patients undergoing EVT through a prospective and retrospective study design. However, a meta-analysis was unable to be performed due to an insufficient number of studies (Table 4).

#### 3.1.5. Prevalence of FR in Asia

Seventeen studies [11,16,22,23,24,30,32,33,38,39,40,43,44,45,46,47,48], comprising a total of 3452 patients, assessed the estimated pooled prevalence of FR amongst AIS patients in Asia (Figure 2). The meta-analysis revealed an estimated pooled prevalence of 53%, ranging from 49% to 58% (ES 53%; 95% CI 49–58%; z = 34.09; *p* < 0.001). Notably, there was a substantial amount of heterogeneity between the studies (I^2^ = 83.73%, *p* < 0.001). The heterogeneity chi^2^ was 98.36 (*p* < 0.001, d.f. 16). The random test for heterogeneity between the subgroups was 19.07 (*p* < 0.001, d.f. 4).

#### 3.1.6. Prevalence of FR in Europe

Ten studies [8,12,14,15,17,21,26,28,34,35], with a cumulative cohort of 4466 patients, assessed the estimated pooled prevalence of FR amongst AIS patients in Europe (Figure 2). The meta-analysis revealed an estimated pooled prevalence of 48%, ranging from 44% to 53% (ES 48%; 95% CI 44–53%; z = 32.84; *p* < 0.01). There was a substantial amount of heterogeneity between the studies (I^2^ = 83.73%, *p* < 0.001). The heterogeneity chi^2^ was 58.04 (*p* < 0.001, d.f. 9).

#### 3.1.7. Prevalence of FR in North America

Four studies [13,25,29,41], encompassing a total of 592 patients, assessed the estimated pooled prevalence of FR amongst AIS patients in North America (Figure 2). The meta-analysis revealed an estimated pooled prevalence of 62%, ranging from 49% to 74% (ES 62%; 95% CI 49–74%; z = 12.68; *p* < 0.001). Notably, there was a substantial amount of heterogeneity between the studies (I^2^ = 89.50%, *p* < 0.001). The heterogeneity chi^2^ was 28.56 (*p* < 0.001, d.f. 5).

#### 3.1.8. Prevalence of FR in the Middle East

Two studies [31,42], with a cohort size of 144 reported on the prevalence of FR in AIS patients undergoing EVT in the Middle East. However, a meta-analysis was unable to be performed due to an insufficient number of studies (Table 4).

#### 3.1.9. Prevalence of FR in Studies Encompassing Data from Multiple Countries

Six studies [18,19,20,36,37,51], with a cumulative cohort of 3049 patients, assessed the estimated pooled prevalence of FR amongst AIS patients in multiple countries (Figure 2). The meta-analysis revealed an estimated pooled prevalence of 50%, ranging from 41% to 59% (ES 50%; 95% CI 41–59%; z = 16.26; *p* < 0.001). Notably, there was a substantial amount of heterogeneity between the studies (I^2^ = 95.34%, *p* < 0.001). The heterogeneity chi^2^ was 107.37 (*p* < 0.001, d.f. 5).

### 3.2. Predictive Indicators of Futile Recanalization

Table 5 and Table 6 summarize the associations between various factors and the likelihood of FR in patients with AIS. For more detailed information on these associations, please refer to the provided text and supplemental figures (Appendix A). Appendix A provide the Egger’s plot for assessing publication bias.

#### 3.2.1. Male Sex

A total of 9848 patients from 34 studies were analyzed [8,12,13,15,16,17,18,19,20,21,22,24,25,26,28,29,30,32,33,34,36,37,38,39,40,41,42,43,44,45,46,47,48,51]. Male sex was associated with decreased odds of FR (OR 0.865; 95% CI: [0.769; 0.973]; *p* = 0.016). This trend was more pronounced in anterior occlusions and prospective studies.

#### 3.2.2. Atrial Fibrillation: (AF)

A total of 28 studies, involving 7471 patients, were included in the analysis of the association between atrial fibrillation and FR [8,15,16,17,18,19,20,21,22,24,25,28,29,30,32,33,34,36,38,39,40,41,43,44,45,46,47,48]. AF was significantly associated with an increased likelihood of FR (OR 1.39; 95% CI: [1.22; 1.59]; *p* < 0.001). A significant association between AF and FR was observed in patients with anterior or mixed occlusions and in both prospective and retrospective study designs.

#### 3.2.3. Alcohol

Eight studies involving 1104 patients were included in the analysis [16,24,33,34,41,43,44,45]. Alcohol was insignificantly associated with an increased likelihood of FR (OR 0.80, 95% CI: [0.58; 1.10]; *p* = 0.170).

#### 3.2.4. Cardiovascular Disease (CVD)

The association between CVD and FR was assessed in 17 studies, involving 2610 patients [16,20,21,24,25,28,32,33,36,38,40,41,42,43,44,45,46,47]. The overall analysis showed that CVD was not significantly associated with an increased risk of FR (OR 1.152, 95% CI: [0.795; 1.671], *p* = 0.454).

#### 3.2.5. Hypertension

In a meta-analysis comprising 33 studies with a total of 9438 patients, HTN was significantly associated with an increased likelihood of FR (OR 1.648, 95% CI: [1.412; 1.924], *p* < 0.001) [8,12,13,15,16,17,18,19,20,21,22,24,25,26,28,29,30,32,33,34,36,38,39,40,41,42,43,44,45,46,47,48,51]. The association between HTN and FR was consistent in patients with anterior circulation occlusions and across prospective and retrospective study designs.

#### 3.2.6. Hyperlipidaemia

In 28 studies, involving 6436 patients, hyperlipidemia (HL) showed no significant association with futile recanalization (FR) in AIS patients undergoing endovascular therapy (OR 0.973, 95% CI: [0.870; 1.088], *p* = 0.627) [12,15,16,17,18,19,20,21,22,24,25,26,28,30,32,34,36,38,39,40,41,42,43,44,45,47,48,51].

#### 3.2.7. Diabetes Mellitus (DM)

The DM and FR meta-analysis involved 33 studies with 9435 patients [8,12,13,15,16,17,18,19,20,21,22,24,25,26,28,29,30,32,33,34,36,38,39,40,41,42,43,44,45,46,47,48,51]. DM was significantly linked to an increased FR risk (OR 1.709, 95% CI: [1.468; 1.990], *p* < 0.001), especially in patients with anterior and mixed occlusions, regardless of study design.

#### 3.2.8. Previous Stroke or Transient Ischemic Attack

In 16 studies with 4820 patients [8,16,24,28,30,32,33,40,41,42,43,44,45,46,48,51], a significant association was found between PS/TIA and FR in AIS patients undergoing EVT. The odds ratio (OR) was 1.298 (95% CI: [1.058; 1.592], *p* = 0.012), indicating an increased likelihood of FR. This association was particularly evident in patients with anterior circulation occlusions and in prospective studies.

#### 3.2.9. Smoking

Smoking was associated with a reduced likelihood of futile recanalization (FR) in 25 studies involving 5595 patients (OR 0.664, 95% CI: [0.572; 0.772], *p* < 0.001) [13,15,16,20,21,22,24,25,26,28,30,32,33,34,39,40,41,42,43,44,45,46,47,48,51]. This association held true for patients with anterior and mixed occlusions, in both prospective and retrospective studies.

#### 3.2.10. Good Collaterals at Baseline

A total of 1925 patients from 7 studies were examined [16,19,30,37,43,48]. GC was significantly associated with decreased odds of FR (OR 0.331, 95% CI: [0.225; 0.486], *p* < 0.001). This trend was observed in anterior occlusions and both prospective and retrospective studies.

#### 3.2.11. Prior Antiplatelet Usage (APU)

A total of 2935 patients from 8 studies were included [12,16,20,24,28,42,46,51]. Prior APU showed no significant association with FR (OR 1.155, 95% CI: [0.976; 1.386], *p* = 0.094).

#### 3.2.12. Prior Anticoagulant Usage (ACU)

A total of 2827 patients from 8 studies were examined [12,16,20,24,28,34,42,51]. Prior ACU was significantly associated with increased odds of FR (OR 1.330, 95% CI: [1.083; 1.634], *p* = 0.007). This association was observed in retrospective studies.

#### 3.2.13. Large-Artery Atherosclerosis (LAA) Etiology

A total of 4083 patients from 15 studies were analyzed [8,12,13,15,16,17,18,19,20,21,22,24,25,26,28,29,30,32,33,34,36,37,38,39,40,41,42,43,44,45,46,47,48,51]. LAA etiology showed no significant association with FR (OR 0.827, 95% CI: [0.671; 1.018], *p* = 0.073).

#### 3.2.14. Cardioembolic (CE) Etiology

A total of 4841 patients from 17 studies were examined [13,16,22,24,26,28,30,33,34,38,42,43,45,46,47,48,51]. CE etiology was significantly associated with increased odds of FR (OR 1.342, 95% CI: [1.100; 1.625], *p* = 0.003). This trend was observed in anterior occlusions and both prospective and retrospective studies.

#### 3.2.15. General Anesthesia (GA)

A total of 5253 patients from 7 studies were included [8,16,18,34,46,48,51]. GA was significantly associated with increased odds of FR (OR 1.533, 95% CI: [1.352; 1.737], *p* < 0.001). This association was observed in anterior/mixed occlusions and both prospective and retrospective studies.

#### 3.2.16. Adjunct Intravenous Thrombolysis (IVT)

A total of 9365 patients from 30 studies were analyzed [8,12,13,15,16,17,18,19,21,26,28,30,32,33,34,36,37,38,39,40,41,42,44,45,46,47,48,51]. Adjunct IVT was significantly associated with decreased odds of FR (OR 0.75 95% CI [0.662; 0.857], *p* < 0.001). This trend was observed in anterior/mixed occlusions and both prospective and retrospective studies.

#### 3.2.17. Age

A total of 7417 patients from 31 studies were examined [12,13,15,16,17,18,19,20,21,22,24,26,28,29,30,32,33,34,36,37,38,39,40,42,44,45,46,47,48,51]. Increasing age was significantly associated with FR (SMD 0.491; 95% CI: [0.417, 0.564], *p* < 0.0001). This association was consistent across occlusion location and study design subgroups.

#### 3.2.18. Baseline Systolic Blood Pressure (SBP)

A total of 4841 patients from 18 studies were included [12,13,20,21,22,24,26,28,29,30,33,36,38,42,44,47,48,51]. Elevating baseline SBP was significantly associated with FR (SMD 0.197; 95% CI: [0.127; 0.266], *p* < 0.001). This trend held for anterior/mixed occlusions and both prospective and retrospective studies.

#### 3.2.19. Baseline National Institute of Health Stroke Severity Score (NIHSS)

A total of 8892 patients from 29 studies were analyzed [8,12,15,16,17,18,19,20,21,22,24,26,28,30,32,33,34,36,38,39,40,42,44,45,46,47,48,51]. Increasing NIHSS score was associated with FR (SMD = 0.753, 95% CI: [0.648; 0.857], *p* < 0.001). This association was consistent across all subgroups.

#### 3.2.20. Baseline Alberta Stroke Program Early CT Score (ASPECTS)

A total of 5588 patients from 22 studies were examined [12,16,17,18,19,20,21,26,28,29,30,32,33,34,38,39,40,42,46,47,48,51]. Decreasing ASPECTS was significantly associated with FR (SMD = −0.368, 95% CI: [−0.464; −0.271], *p* < 0.001). This trend held for anterior/mixed occlusions and both prospective and retrospective studies.

#### 3.2.21. Onset-to-Treatment Time (OTT)

A total of 6964 patients from 18 studies were included [8,12,15,17,18,19,20,22,24,26,28,30,32,36,38,40,46,47,51]. Longer OTT was significantly associated with FR (SMD = 0.466, 95% CI: [0.364; 0.569], *p* < 0.001). This association was consistent across occlusion location and study design subgroups.

#### 3.2.22. Onset-to-Reperfusion Time (OTR)

A total of 2716 patients from 17 studies were included [12,15,16,17,18,24,26,32,34,36,39,42,43,45,47]. Longer OTR was significantly associated with FR (SMD = 0.375, 95% CI: [0.185; 0.565], *p* < 0.001). This was consistent across all subgroups.

#### 3.2.23. Baseline Blood Glucose

Baseline blood glucose (BG) was significantly associated with FR in AIS patients undergoing endovascular therapy in 17 studies with 5099 patients (SMD = 0.313, 95% CI: [0.217; 0.409], *p* < 0.001) [12,14,16,20,24,26,28,30,32,33,38,43,45,46,47,48,51]. Higher BG levels correlated with an increased likelihood of FR. This association was consistent for anterior and mixed occlusions and across study designs.

### 3.3. Clinical Outcomes Following Futile Recanalization

In the analysis of clinical outcomes following FR, we investigated three key factors (Figure 3 and Figure 4). First, our meta-analysis, involving 17 studies with a total of 4634 patients [8,17,18,22,24,25,28,29,30,34,36,37,38,39,42,45,47], demonstrated a significant association between sICH and FR, with an OR of 7.372 (95% CI: [4.889; 11.116], *p* < 0.001). This relationship held across various subgroups, including occlusion location and study design, with low heterogeneity observed (I^2^ = 0.0%, *p* = 0.511). Second, when examining HT in 11 studies comprising 2098 patients [19,26,29,30,32,33,37] we found a notable connection between FR and an increased likelihood of HT (OR 2.982, 95% CI: [2.374; 3.746], *p* < 0.001), consistent across occlusion locations (Figure 3) and study designs (Figure 4), with low heterogeneity (I^2^ = 4.7%, *p* = 0.398). Third, in the assessment of mortality across five studies with a total of 934 patients [19,20,30,39,42], we observed a significant association between FR and increased odds of mortality (OR 19.235, 95% CI [1.573; 235.178], *p* = 0.021), albeit with substantial heterogeneity (I^2^ = 78.2%, *p* < 0.001). While these findings offer valuable insights into the implications of FR among AIS patients, it is important to note that some evidence of publication bias was observed. Detailed data are presented in Table 5, Appendix A, and Appendix A, which provide a comprehensive view of the meta-analysis outcomes, further stratified by occlusion location and study design. Analyses of publication bias through Egger’s plot and Egger’s test are provided in Appendix A and Appendix A.

## 4. Discussion

Our meta-analysis has revealed a pooled prevalence estimate of 51% for FR among AIS patients undergoing EVT. This study is distinct in providing pooled prevalence estimates on FR following EVT for AIS patients, presenting the largest sample size reported to date. Additionally, we have also identified clinical risk factors significantly linked to FR, including age, AF, HTN, DM, history of stroke and/or transient ischemic attack, and smoking. Furthermore, FR is correlated with an elevated risk of severe adverse outcomes, encompassing sICH, HT, and mortality.

We have determined a pooled prevalence of 51% for FR among patients with AIS who undergo EVT. This finding contrasts with earlier meta-analyses, which reported prevalence rates ranging from 32.4% to 56.7% [3,7]. Establishing this pooled prevalence of FR is pivotal in recognizing and conveying the potential risks associated with undergoing EVT for AIS patients. The notable heterogeneity observed across the studies could be attributed to procedural disparities among treatment centers, FR in the timing of patient presentations leading to differences in OTT and OTR. Furthermore, the divergence in hospital settings and statuses, particularly between tertiary hospitals and other centers, could contribute to the observed heterogeneity. Tertiary hospitals, likely catering to patients with more severe stroke symptoms, might yield outcomes distinct from centers treating milder symptoms [52,53]. The occurrence of FR might be explained by the no-reflow phenomenon. This phenomenon arises from leukocyte-endothelial interactions that depend on adhesive molecules, leading to the aggregation of red blood cells and the formation of microthrombi [6]. Other factors contributing to this occurrence include early re-occlusion of arteries, hemorrhagic transformation, and compromised collateral circulation [6,54].

Our comprehensive meta-analysis has revealed a spectrum of factors associated with FR following AIS providing critical insights into the multifaceted nature of this phenomenon. These factors encompass a wide range of characteristics, including patient demographics, clinical variables, treatment modalities, and even laboratory markers. Notably, male gender, smoking, the presence of good collaterals, AF, hypertension, diabetes mellitus, prior stroke or TIA, prior use of anticoagulants, cardioembolic etiology, general anesthesia administration, and adjunct IVT are associated with an increased risk of FR. Furthermore, increasing age, systolic blood pressure, NIHSS at admission, onset-to-treatment time, onset-to-reperfusion time, baseline blood glucose, and reduced ASPECTS were correlated with FR [3]. These predictors can help inform clinical decision making and highlight areas for further research. Male sex demonstrated a decreased odd of FR, suggesting that female AIS patients may face a slightly higher risk of FR. AF was significantly associated with increased odds of FR, underscoring the importance of managing AF in AIS patients to improve recanalization outcomes. Hypertension significantly increased the odds of FR, highlighting the need for aggressive blood pressure control in these cases. Conversely, smoking was significantly associated with decreased odds of FR, suggesting a potential protective effect. Furthermore, adjunct intravenous thrombolysis significantly decreased the odds of FR, emphasizing the importance of considering IVT in AIS management. However, the study revealed several important nuances. For instance, certain factors, like alcohol and hyperlipidemia, showed no significant association with FR, suggesting that their impact on recanalization outcomes may be limited. These findings are in line with previous meta-analyses that have identified various factors linked to FR, including age, admission NIHSS score, ASPECTS score, HTN, admission SBP, AF, and the usage of intravenous tissue plasminogen activator (IV tPA), OTT, and OTR [3,7]. However, our analysis adds to the existing literature by providing a more comprehensive overview of the factors associated with FR, encompassing a broader range of variables.

It is important to acknowledge that the included studies varied in terms of patient populations, methodologies, and reporting standards. Factors such as race, education levels, body mass index (BMI), wake-up stroke, statin usage, and comorbidities like congestive heart failure and intracranial atherosclerotic stenosis were inconsistently reported or underrepresented [14,20,22,24,25,30,36,47,48,51]. Laboratory markers such as leukocyte status [12,32,45], high-sensitivity c-reactive protein (hs-CRP) levels [26,38,42], and various blood cell counts were also underreported [12,16,20,28,38,42,44]. Imaging characteristics, stroke location, etiology, and procedural details demonstrated significant heterogeneity across studies. These variations underscore the complexity of FR and emphasize the need for standardized reporting and further research to better understand the interplay of these factors in clinical practice. Additionally, the study detected publication bias in several analyses, underscoring the need for further research and cautious interpretation of these findings. Overall, these results indicate that risk factors associated with FR following AIS are diverse and multifactorial, encompassing patient characteristics, clinical variables, and treatment-related factors. While this meta-analysis sheds light on many of these factors, it also highlights the need for more comprehensive and standardized research to enhance our understanding of FR and improve patient outcomes. Additionally, considering the multifaceted nature of FR, a personalized approach to stroke management, taking into account these various factors, may be necessary to reduce the incidence of FR and optimize stroke care.

Our meta-analysis has uncovered a significant link between FR and unfavorable outcomes encompassing sICH, HT, and mortality. This is particularly concerning given that even within the EVT subgroup, certain patients continue to encounter adverse long-term clinical outcomes, despite achieving optimal recanalization rates. While numerous studies have investigated the relationship between FR and diverse adverse outcomes, our ability to conduct a comprehensive meta-analysis was constrained by the limited number of available studies for each specific outcome. These outcomes encompass a wide spectrum, ranging from early neurological deterioration [20,47] and neurologic progression [22] to brain herniation [47], parenchymal hypodensities or local brain swelling [47], hemorrhagic infarction [12], subarachnoid hemorrhage [12,48], parenchymal hematomas [12,26], device or procedural complications [14,20,36], 24 h NIHSS changes [14,19], duration of intensive care unit (ICU) stay [20], intubation within 7 h of stroke onset [20], or intubation [36], hospitalization duration [20], trail making at 90 days [20], distal clot migration [29], and asymptomatic intracranial hemorrhage [20,39]. Moreover, post-procedural factors such as discharge destination have also emerged as outcomes associated with FR [20]. Similarly, imaging characteristics such as initial infarct volume [12,19], final infarct volume [40,48], ischemic core volume [19,30,40], hypoperfusion volume [30], extent of mismatch [30], CHA_2_DS_2_-VASc scores [42], and large deep white lesions [40] were not adequately reported. Details regarding stroke location and etiology (e.g., vertebral or basilar artery [24,36], anterior carotid artery [44], carotid terminus [37], posterior artery [42]), or extracranial stroke [48]) were also lacking. Furthermore, in-hospital measurements and procedural disparities, such as baseline AOL grade [33], angioplasty procedure [32], IVT dosage [20], onset-to-IVT-dosage time [20], rescue therapy [30,36], type of IVT administered [48], hospital transfer [27], and number of passes used [43]), varied across studies. Different devices were employed, including MERCI retriever [13], intracranial stent [13], penumbra reperfusion systems [39] and multiple devices [39]. Differing parameters of reporting pre-stroke mRS scores were also evident, with some studies reporting mRS scores as 0 or 1–2 [20], others as 0–1 or 2–3 [43], and some indicating only patient numbers with an mRS > 2 [26].

## 5. Limitations

Our meta-analysis exhibits certain limitations arising from variations in the quality of the included studies, which in turn impacted our precision in data extraction and analysis. Initially, we encountered a range of study designs, spanning from retrospective designs to prospective approaches and randomized controlled trials (RCTs). Furthermore, the research objectives differed; some studies aimed to contrast the outcomes of distinct treatment regimens (such as comparing EVT solely to EVT and IVT), while others focused on identifying potential adverse consequences of EVT. Additionally, a considerable portion of the studies was retrospective in nature, consequently constraining their overall design.

Additionally, it is worth acknowledging the potential impact of the 2015 American Heart Association (AHA)/American Stroke Association (ASA) guideline changes [55], which could introduce variations in results between studies conducted before and after that year. Looking ahead, we propose further investigation into the prevalence of FR among AIS patients, with a specific focus on regional disparities. These forthcoming studies should also delve into the various associated factors with FR, particularly those that have been identified as underreported, thus precluding a comprehensive meta-analysis. An essential consideration would be to establish long-term clinical outcomes as primary endpoints for RCTs. This approach would enable accurate identification and analysis of adverse outcomes stemming from FR and facilitate in-depth scrutiny of factors that could potentially influence FR occurrences. Addressing the existing data gaps, such as collateral status and differences in EVT devices, should also be part of the analytical strategy. To ensure uniformity in research findings, harmonizing the definitions of FR across all centers is paramount. Furthermore, optimizing systems-level factors like time to treatment or procedural duration across centers and regions could enhance the reliability and applicability of findings [53,56]. Recently, machine learning-based models have been tested to preoperatively predict the occurrence of FR [9]. Further work is required to evaluate these algorithms and their accuracy in real-world settings.

## 6. Conclusions

In summary, our study has revealed an estimated pooled prevalence of 51% of FR among patients undergoing EVT for AIS. Our investigation has successfully identified several factors associated with FR, including male gender, smoking, good collaterals, atrial fibrillation, hypertension, diabetes, previous history of stroke/TIA, anticoagulant use, cardioembolism etiology, general anesthesia administration, adjunct intravenous thrombolysis, increasing age, systolic blood pressure, NIHSS score, onset-to-treatment time, onset-to-reperfusion time, baseline glucose, and ASPECTS reduction. Additionally, we have identified significant outcomes linked to FR, specifically sICH, HT, and 90-day mortality. Further research is warranted to delve deeper into potential additional associated factors and outcomes following FR. This pursuit will enable the identification of high-risk patients and enable the formulation of implementation of efficacious recommendations to optimize their care.

## Figures and Tables

**Figure 1 life-13-01965-f001:**
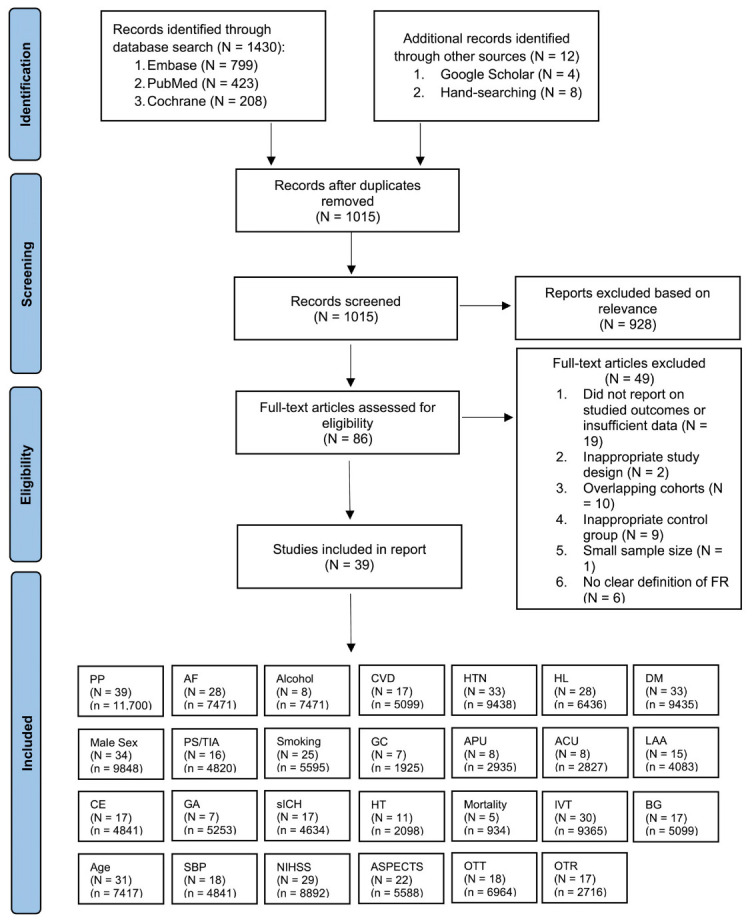
PRISMA Flowchart: Inclusion of Studies in the Meta-Analysis. Illustration depicting the flow of study selection according to the PRISMA guidelines, leading to the inclusion of studies in the meta-analysis. Abbreviations: N: number of studies; n: cohort size; PP = pooled prevalence; AF: atrial fibrillation, CVD: cardiovascular disease; HTN: hypertension; HL: hyperlipidemia; DM: diabetes mellitus; PS/TIA: previous stroke/transient ischemic attack; GC: good collaterals; APU: antiplatelet usage; ACU: anticoagulant usage; LAA: large-artery atherosclerosis; CE: cardioembolic; IVT: intravenous thrombolysis; GA: general anesthesia; sICH: symptomatic intracranial hemorrhage; HT: hemorrhagic transformation; BG: blood glucose; SBP: systolic blood pressure; NIHSS: baseline National Institute of Health Stroke Severity; ASPECTS: Alberta Stroke Program Early CT Score; OTT: onset-to-treatment time; OTR: onset-to-recanalization time.

**Figure 2 life-13-01965-f002:**
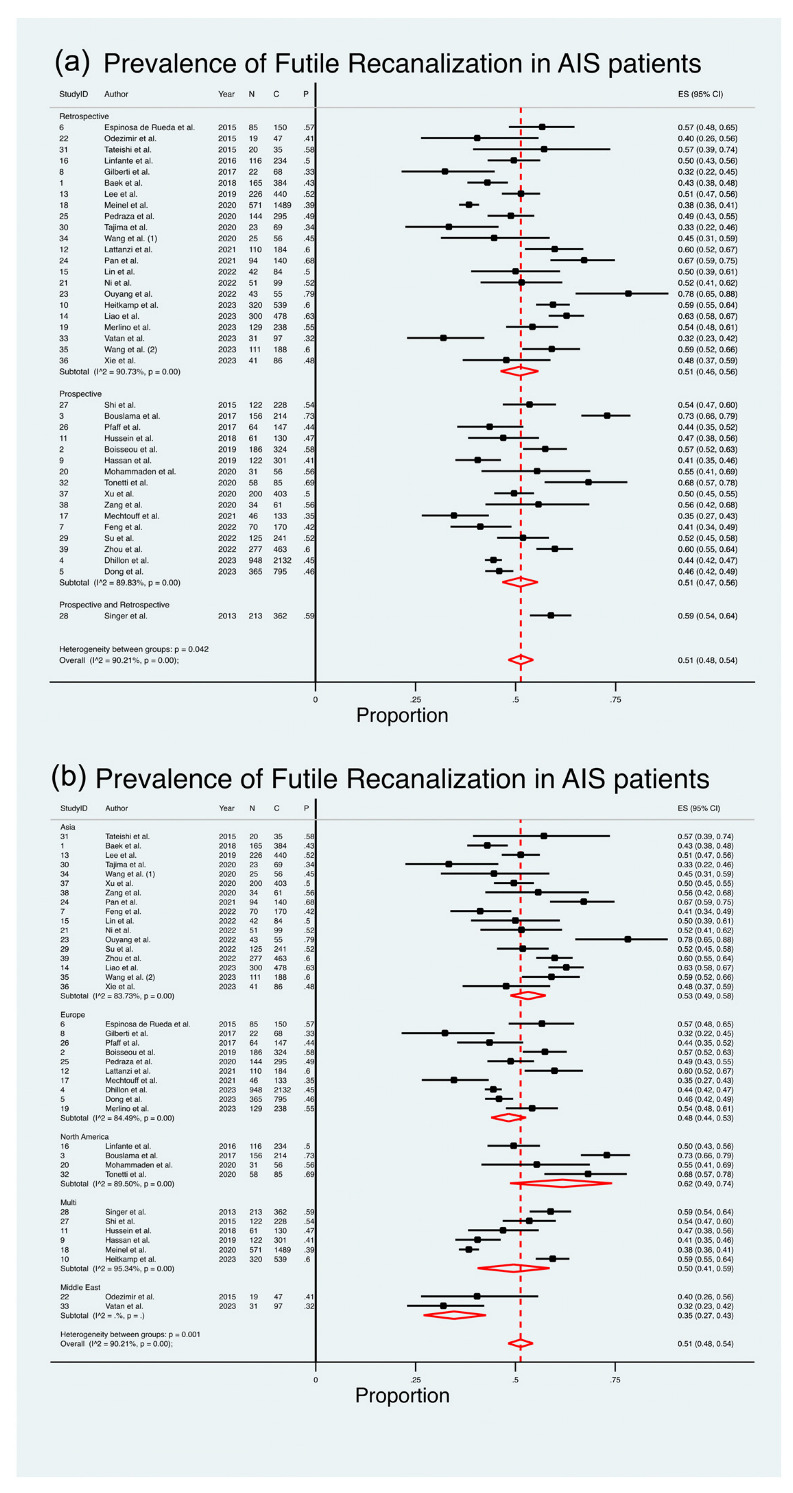
Forest Plot: Pooled Prevalence of Futile Recanalization—Stratified by Study Design and Geographical Region [8,11,12,13,14,15,16,17,18,19,20,21,22,23,24,25,26,28,29,30,31,32,33,34,35,36,37,38,39,40,41,42,43,44,45,46,51]. Abbreviations: AIS: acute ischemic stroke; CI: confidence interval; DL: DerSimonian and Laird method.

**Figure 3 life-13-01965-f003:**
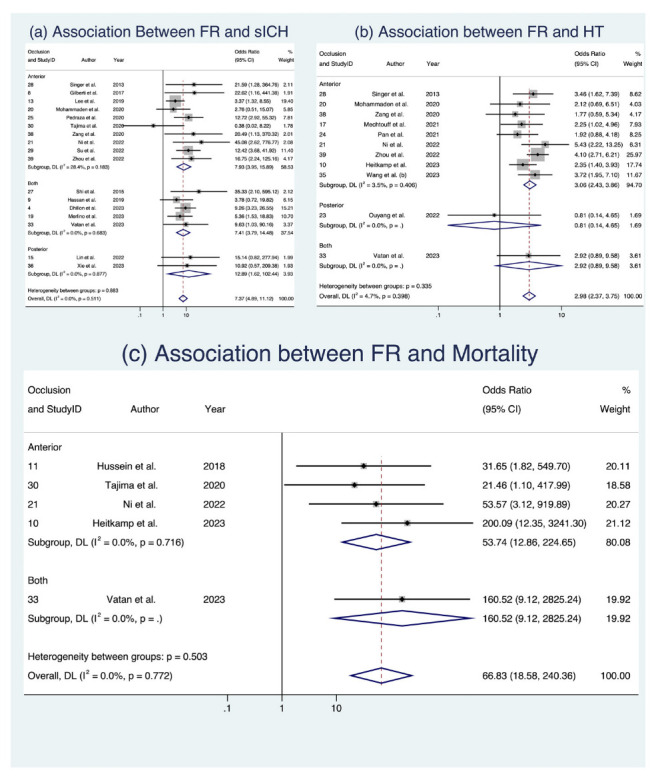
Forest plots: clinical outcome analysis of futile recanalization—stratified by occlusion location [8,17,18,22,24,25,28,29,30,34,36,37,38,39,42,45,47]. Abbreviations: FR: futile recanalization; sICH: symptomatic intracranial hemorrhage; HT: hemorrhagic transformation; CI: confidence interval; DL: DerSimonian and Laird method.

**Figure 4 life-13-01965-f004:**
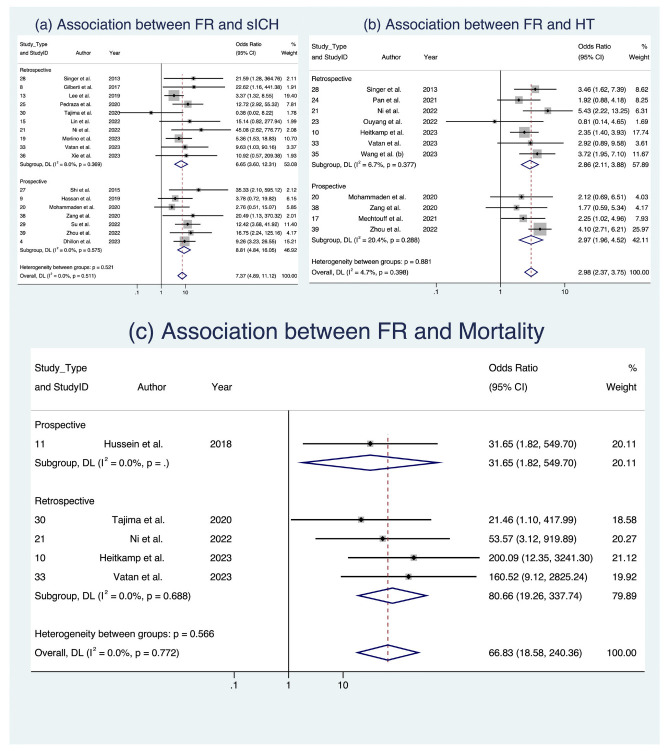
Forest plots: clinical outcome analysis of futile recanalization—stratified by study design [8,17,18,22,24,25,28,29,30,34,36,37,38,39,42,45,47]. Abbreviations: FR: futile recanalization; sICH: symptomatic intracranial hemorrhage; HT: hemorrhagic transformation; CI: confidence interval; DL: DerSimonian and Laird method.

**Table 1 life-13-01965-t001:** Clinical Characteristics of Studies Included in Meta-Analysis.

Author	Year	Country	Study Design	Cohort Size	Age (±SD)	Male (n%)	Occlusion Location	Reperfusion	FR (n%)	Poor Outcome Criteria	Recanalization Criteria	Etiology Criteria	Collateral Criteria	sICH Criteria	sICH		HT		Mortality	
															FR, n (n%)		FR, n (n%)		FR, n (n%)	
															Yes	No	Yes	No	Yes	No
Baek et al. [11]	2018	South Korea	Retrospective	384	67.3 (±12.4)	-	Anterior	EVT ± IVT	43%	mRS 3–6 at 90 days	mTICI 2b-3	-	-	-	-	-	-	-	-	-
Boisseou et al. [12]	2019	France	Prospective	324	64 (±2.98)	51%	Anterior	EVT ± IVT	57%	mRS 3–6 at 90 days	mTICI 2b-3	-	-	-	-	-	-	-	-	-
Bouslama et al. [13]	2017	USA	Prospective	214	64.3 (±13.5)	54%	Posterior	EVT ± IVT	73%	mRS 3–6 at 90 days	mTICI 2b-3	-	-	-	-	-	-	-	-	-
Dhillon et al. [8]	2023	United Kingdom	Prospective	2132	-	56%	Both	EVT ± IVT	44%	mRS 4–6 at discharge	mTICI 3	-	-	ECASS II	28 (4.40)	4 (0.49)	-	-	-	-
Dong et al. [14]	2023	France	Prospective	795	-	-	Anterior	EVT ± IVT	46%	mRS 3–6 at 90 days	mTICI 3	TOAST	ASITN/SIR	-	-	-	-	-	-	-
Espinosa de Rueda et al. [15]	2015	Spain	Retrospective	150	-	52%	Anterior	EVT ± IVT	57%	mRS 3–6 at 90 days	TICI 2b-3	-	CTA-maximum intensity projection images	-	-	-	-	-	-	-
Feng et al. [16]	2022	China	Prospective	170	66.37 (±11.59)	68%	Anterior	EVT ± IVT	41%	mRS 3–6 at 90 days	mTICI 3	TOAST	ASITN/SIR	-	-	-	-	-	-	-
Gilberti et al. [17]	2017	Italy	Retrospective	68	73 (±9.85)	50%	Anterior	EVT ± IVT	32%	mRS >2 at 90 days	TICI 2b-3	TOAST	-	ECASS II	4 (18.18)	0 (0.0)	-	-	-	-
Hassan et al. [18]	2019	Multi	Prospective	301	-	41%	Both	EVT ± IVT	41%	mRS 3–6 at 90 days	TICI 2b-3	-	-	NIHSS	5 (4.09)	2 (1.11)	-	-	-	-
Heitkamp et al. [19]	2023	Multi	Retrospective	539	73.67 (±14.87)	48%	Anterior	EVT ± IVT	59%	mRS 3–6 at 90 days	mTICI 2b-3	-	Tan Scale	-	-	-	72 (26.47)	23 (13.29)	100 (31.25)	0 (0.0)
Hussein et al. [20]	2018	Multi	Prospective	130	65.18 (±12.6)	48%	Anterior	EVT ± IVT	47%	mRS 3–6 at 90 days	mTICI 2b-3	-	-	-	-	-	-	-	11 (18.03)	0 (0.0)
Lattanzi et al. [21]	2021	Italy	Retrospective	184	72.34 (14.95)	48%	Anterior	EVT ± IVT	60%	mRS 3–6 at 90 days	mTICI 2b-3	-	-	-	-	-	-	-	-	-
Lee et al. [22]	2019	South Korea	Retrospective	440	67.3 (±12.3)	58%	Anterior	EVT ± IVT	51%	mRS 3–6 at 90 days	TICI 2b-3	TOAST	-		20 (8.85)	6 (2.80)	-	-	-	-
Liao et al. [23]	2023	China	Retrospective	478	-	-	Posterior	EVT ± IVT	63%	mRS 4–6 at 90 days	mTICI 2b-3	-	-		-	-	-	-	-	-
Lin et al. [24]	2022	China	Retrospective	84	65.87 (±12.04)	81%	Posterior	EVT ± IVT	50%	mRS 3–6 at 90 days	TICI 2b-3	TOAST	-	-	6 (14.3)	0 (0.0)	-	-	0 (0.0)	9 (21.43)
Linfante et al. [25]	2016	USA	Retrospective	234	-	49%	Both	EVT ± IVT	50%	mRS 3–6 at 90 days	TICI 2b-3	-	-	-	-	-	-	-	-	-
Mechtouff et al. [26]	2021	France	Prospective	133	-	61%	Anterior	EVT ± IVT	35%	mRS 3–6 at 90 days	TICI 2b-3	-	Hishigada Score	-	-	-	17 (37.00)	18 (20.69)	-	-
Meinel et al. [27]	2020	Multi	Retrospective	1489	-	52%	Both	EVT ± IVT	38%	mRS 4–6 at 90 days	mTICI 2b-3	TOAST	-	ECASSII	-	-	-	-	-	-
Merlino et al. [28]	2023	Italy	Retrospective	238	-	48%	Both	EVT ± IVT	54%	mRS 3–6 at 90 days	TICI 2b-3	TOAST	-	NIHSS	17 (13.12)	3 (2.75)	-	-	-	-
Mohammaden et al. [29]	2020	USA	Prospective	56	-	52%	Anterior	EVT	55%	-	6	mRS 3–6 at 90 days	mTICI 2b-3	-	6 (19.35)	2 (8.00)	14 (45.16)	7 (28.00)	-	-
Ni et al. [30]	2022	China	Retrospective	99	72.67 (±10.54)	52%	Anterior	EVT ± IVT	52%	mRS 3–6 at 90 days	mTICI 2b-3	-	ASITN/SIR	NIHSS	16 (31.37)	0 (0.0)	30 (58.82)	10 (20.83)	18 (35.29)	0 (0.0)
Odezimir et al. [31]	2015	Turkey	Retrospective	47	-	50%	Both	EVT ± IVT	40%	mRS 3–6 at 90 days	TICI 2b-3	-	-	-	-	-	-	-	-	-
Ouyang et al. [32]	2022	China	Retrospective	55	65.34 (±13.71)	-	Posterior	EVT ± IVT	78%	mRS 3–6 at 90 days	mTICI 2b-3	-	-	-	-	-	6 (13.95)	2 (16.67)	-	-
Pan et al. [33]	2021	China	Retrospective	140	68.17 (±14.24)	86%	Anterior	EVT ± IVT	67%	mRS 3–6 at 90 days	mTICI 2b-3	TOAST	-	-	-	-	38 (40.43)	12 (26.09)	-	-
Pedraza et al. [34]	2020	Spain	Retrospective	295	71.29	61%	Anterior	EVT ± IVT	48%	mRS 3–6 at 90 days	mTICI 2b-3	TOAST	-	ECASS II	21 (14.58)	2 (1.32)	-	-	-	-
Pfaff et al. [35]	2017	Germany	Prospective	147	-	54%	Anterior	EVT ± IVT	44%	mRS 4–6 at 90 days	TICI 2b-3	-	-	ECASS II	-	-	-	-	-	-
Shi et al. [36]	2015	Multi	Prospective	228	-	-	Both	EVT ± IVT	54%	mRS 3–6 at 90 days	TICI 2b-3	-	-	NIHSS	17 (13.93)	0 (0.0)	-	-	-	-
Singer et al. [37]	2013	Multi	Retrospective	362	66.67 (±14.89)	44%	Anterior	EVT ± IVT	59%	mRS 3–6 at 90 days	TIMI 2 or 3	-	-	ECASSII	14 (6.57)	0 (0.0)	39 (18.31)	9 (6.10)	-	-
Su et al. [38]	2022	China	Prospective	241	70.29 (±11.77)	-	Anterior	EVT ± IVT	52%	mRS 3–6 at 90 days	mTICI 2b-3	-	ASITN/SIR	ECASS II	31 (24.8)	3 (2.59)	-	-	-	-
Tajima et al. [39]	2020	Japan	Retrospective	69	74.6 (±9.2)	50%	Anterior	EVT ± IVT	34%	mRS 3–6 at 90 days	mTICI 2b-3	-	-	NIHSS	0 (0.0)	2 (4.34)	-	-	4 (17.39)	0 (0.0)
Tateishi et al. [40]	2015	Japan	Retrospective	35	68.34 (±16.24)	58%	Anterior	EVT ± IVT	57%	mRS 3–6 at 90 days	TICI 2b-3	-	-	-	-	-	-	-	-	-
Tonetti et al. [41]	2020	USA	Prospective	85	71.1 (±16.00)	52%	Anterior	EVT ± IVT	68%	mRS 4–6 at 90 days	mTICI 2b-3	-	-	-	-	-	-	-	-	-
Vatan et al. [42]	2023	Turkey	Retrospective	97	66.5 (±12)	51%	Both	EVT ± IVT	32%	mRS 3–6 at 90 days	mTICI 2c-3	TOAST	TAN Grading System	NIHSS	4 (12.90)	1 (1.51)	7 (22.58)	6 (9.09)	17 (54.84)	0 (0.0)
Wang et al. (1) [43]	2020	China	Retrospective	56	65.2 (±5.2)	50%	Anterior	EVT	45%	mRS 3–6 at 90 days	mTICI 2b-3	-	ASITN/SIR	-	-	-	-	-	-	-
Wang et al. (2) [44]	2023	China	Retrospective	188	69 (±12.71)	70%	Anterior	EVT ± IVT	59%	mRS 3–6 at 90 days	mTICI 2b-3	TOAST	-	-	-	-	59 (53.15)	18 (23.38)	-	-
Xie et al. [45]	2023	China	Retrospective	86	-	61%	Posterior	EVT ± IVT	48%	mRS >3 at 90 days	eTICI 2b-3	TOAST	-	-	4 (9.76)	0 (0.0)	-	-	-	-
Xu et al. [46]	2020	China	Prospective	403	64.2 (±13.9)	80%	Anterior	EVT ± IVT	50%	mRS 3–6 at 90 days	mTICI 2b-3	-	-	-	-	-	-	-	-	-
Zang et al. [47]	2020	China	Prospective	61	-	62%	Anterior	EVT ± IVT	56%	mRS 3–6 at 90 days	mTICI 2b-3	TOAST	-	NIHSS	9 (2.64)	0 (0.0)	13 (38.24)	7 (25.93)	-	-
Zhou et al. [48]	2022	China	Prospective	463	68.67 (±11.16)	69%	Anterior	EVT ± IVT	60%	mRS 3–6 at 90 days	eTICI 2b-3	-	Pial Arterial Filling Score	-	23 (8.3)	1 (0.53)	153 (55.23)	43 (23.12)	-	-

Abbreviations: SD: standard deviation; n: number of patients; EVT: endovascular thrombectomy; IVT: intravenous thrombolysis; mRS: modified Rankin scale; TICI: treatment in cerebral infarction; mTICI: modified treatment in cerebral infarction; eTICI: electronic treatment in cerebral infarction; NIHSS: National Institute of Health Stroke Severity; TOAST: Trial of Org 10172 in Acute Stroke Treatment; ASTIN/SIR: American Society of Interventional and Therapeutic Neuroradiology/Society of Interventional Radiology; ECASS II: European Co-operative Acute Stroke Study-II.

**Table 2 life-13-01965-t002:** Discrete Predictive Markers of Futile Recanalization Included in Meta-Analysis.

Study	Male		AF		Alcohol		CVD		HTN		HL		DM		PS/TIA		Smoking		GC		APU		ACU		LAA		CE		GA		IVT	
	FR, n (n%)		FR, n (n%)		FR, n (n%)		FR, n (n%)		FR, n (n%)		FR, n (n%)		FR, n (n%)		FR, n (n%)		FR, n (n%)		FR, n (n%)		FR, n (n%)		FR, n (n%)		FR, n (n%)		FR, n (n%)		FR, n (n%)		FR, n (n%)	
	Yes	No	Yes	No	Yes	No	Yes	No	Yes	No	Yes	No	Yes	No	Yes	No	Yes	No	Yes	No	Yes	No	Yes	No	Yes	No	Yes	No	Yes	No	Yes	No
Baek et al. [11]	-	-	-	-	-	-	-	-	-	-	-	-	-	-	-	-	-	-	-	-	-	-	-	-	-	-	-	-	-	-	-	-
Boisseou et al. [12]	89 (47.85)	71 (51.45)	-	-	-	-	-	-	121 (65.05)	70 (50.73)	52 (27.96)	36 (26.09)	47 (25.27)	17 (12.32)	-	-	-	-	-	-	46 (24.73)	36 (26.09)	41 (22.04)	26 (18.84)	-	-	-	-	-	-	106 (56.99)	84 (60.87)
Bouslama et al. [13]	81 (51.92)	34 (58.62)	-	-	-	-	-	-	115 (73.72)	36 (62.10)	-	-	38 (24.40)	11 (19.00)	-	-	33 (21.15)	26 (44.83)	-	-	-	-	-	-	45 (28.85)	16 (27.59)	45 (28.85)	20 (34.48)	-	-	49 (31.41)	20 (34.48)
Dhillon et al. [8]	527 (55.59)	665 (56.17)	249 (26.27)	223 (18.83)	-	-	-	-	482 (50.84)	527 (44.51)	-	-	164 (17.30)	137 (11.57)	162 (17.09)	179 (15.12)	-	-	-	-	-	-	-	-	-	-	-	-	575 (60.66)	563 (47.55)	505 (53.27)	772 (65.20)
Dong et al. [14]	-	-	-	-	-	-	70 (19.18)	70 (16.28)	-	-	-	-	69 (18.91)	59 (13.72)	-	-	-	-	-	-	-	-	79 (21.64)	55 (12.79)	-	-	-	-	-	-	-	-
Espinosa de Rueda et al. [15]	44 (51.77)	33 (50.77)	36 (42.35)	32 (49.23)	-	-	-	-	63 (74.12)	37 (56.92)	36 (42.35)	32 (49.23)	26 (30.59)	13 (20.00	-	-	21 (24.71)	20 (30.77)	-	-	-	-	-	-	-	-	-	-	-	-	39 (45.88)	28 (43.08)
Feng et al. [16]	47 (67.14)	68 (68.00)	20 (28.57)	27 (27)	22 (31.43)	36 (36.00)	24 (34.29)	18 (18.00)	50 (71.43)	61 (61.00)	20 (28.57)	23 (23.00)	27 (38.57)	22 (22.00)	17 (24.29)	14 (14.00)	27 (38.57)	36 (36.00)	13 (18.57)	57 (57.00)	19 (27.14)	17 (17.00)	10 (14.29)	5 (5.00)	22 (31.43)	44 (44.00)	46 (65.72)	54 (54.00)	20 (28.57)	15 (15.00)	25 (35.72)	37 (37.00)
Gilberti et al. [17]	13 (59.09)	21 (45.65)	12 (54.55)	24 (52.17)	-	-	-	-	16 (72.73)	33 (71.74)	7 (31.82)	16 (34.78)	6 (27.27)	4 (8.70)	-	-	-	-	-	-	-	-	-	-	-	-	-	-	-	-	6 (27.27)	27 (58.70)
Hassan et al. [18]	53 (43.44)	75 (41.90)	50 (40.98)	69 (38.55)	-	-	-	-	76 (62.30)	117 (65.36)	34 (27.87)	62 (34.64)	28 (22.95)	26 (14.53)	-	-	-	-	-	-	-	-	-	-	-	-	-	-	79 (64.76)	103 (57.54)	73 (59.84)	125 (69.83)
Heitkamp et al. [19]	139 (43.44)	119 (54.34)	138 (43.53)	83 (38.07)	-	-	-	-	228 (71.70)	139 (63.47)	81 (28.93)	62 (30.54)	74 (23.34)	35 (15.98)	-	-	-	-	182/316	185/219	-	-	-	-	-	-	-	-	-	-	143 (45.40)	125 (57.60)
Hussein et al. [20]	23 (37.71)	39 (56.52)	27 (44.26)	24 (34.78)	-	-	17 (27.87)	9 (13.04)	44 (72.13)	44 (63.77)	25 (40.98)	32 (46.38)	18 (29.51)	6 (8.70)	-	-	15 (24.59)	19 (27.54)	-	-	25 (40.98)	31 (44.93)	19 (31.15)	24 (34.78)	-	-	-	-	-	-	-	-
Lattanzi et al. [21]	48 (43.64)	39 (52.70)	-	-	-	-	17 (15.46)	17 (22.97)	79 (71.82)	38 (51.35)	48 (43.64)	36 (48.65)	17 (15.46)	6 (8.12)	-	-	22 (20.00)	17 (22.97)	-	-	-	-	-	-	-	-	-	-	-	-	60 (54.55)	50 (67.57)
Lee et al. [22]	120 (53.10)	135 (63.09)	120 (53.10)	108 (50.47)	-	-	-	-	151 (66.82)	122 (57.01)	46 (20.35)	47 (21.96)	58 (25.66)	42 (19.63)	-	-	37 (16.37)	62 (28.97)	-	-	-	-	-	-	51 (22.57)	50 (23.37)	131 (57.97)	121 (56.74)	-	-	154 (68.14)	154 (71.96)
Liao et al. [23]	-	-	-	-	-	-	71 (23.67)	21 (11.80)	-	-	-	-	-	-	-	-	-	-	-	-	-	-	-	-	-	-	-	-	-	-	-	-
Lin et al. [24]	36 (85.72)	32 (76.19)	10 (23.81)	10 (23.81)	14 (33.34)	18 (42.86)	5 (11.91)	6 (14.29)	33 (78.57)	32 (76.19)	18 (42.86)	17 (40.48)	13 (30.95)	14 (33.34)	16 (38.10)	5 (11.91)	26 (61.91)	24 (57.14)	-	-	3 (7.14)	4 (9.52)	2 (4.76)	3 (7.14)	29 (69.05)	32 (76.19)	11 (26.19)	5 (11.91)	-	-	14 (33.34)	18 (42.86)
Linfante et al. [25]	62 (53.50)	52 (44.07)	52 (44.83)	46 (39.10)	-	-	42 (36.21)	34 (28.81)	89 (76.73)	85 (72.03)	63 (54.31)	61 (51.70)	22 (18.97)	39 (33.05)	-	-	28 (24.56)	39 (33.34)	-	-	-	-	-	-	-	-	-	-	-	-	-	-
Mechtouff et al. [26]	25 (54.35)	55 (63.22)	-	-	-	-	-	-	36 (78.26)	31 (35.63)	16 (34.78)	21 (24.14)	13 (28.26)	10 (11.50)	-	-	4 (8.70)	23 (26.44)	-	-	-	-	-	-	9 (19.57)	10 (11.50)	25 (54.35)	45 (51.73)	-	-	19 (41.31)	54 (62.07)
Meinel et al. [27]	292 (51.14)	480 (52.29)	-	-	-	-	-	-	388 (67.95)	572 (62.31)	273 (47.81)	459 (50.00)	120 (21.02)	116 (12.64)	87 (15.24)	105 (11.44)	135 (23.64)	277 (30.18)	-	-	174 (30.47)	242 (26.36)	92 (16.11)	118 (12.86)	81 (14.19)	128 (13.94)	279 (48.86)	415 (45.21)	330 (57.79)	458 (49.89)	253 (44.31)	467 (50.87)
Merlino et al. [28]	53 (41.09)	59 (54.13)	41 (31.78)	24 (22.02)	-	-	16 (12.40)	19 (17.43)	94 (72.87)	67 (61.47)	34 (26.36)	28 (25.69)	19 (14.73)	14 (12.85)	12 (9.30)	13 (11.93)	15 (11.63)	21 (19.27)	-	-	32 (24.81)	27 (24.77)	25 (19.38)	15 (13.76)	13 (10.08)	17 (15.60)	76 (58.92)	58 (53.21)	-	-	67 (51.94)	71 (65.14)
Mohammaden et al. [29]	14 (45.16)	15 (60.00)	12 (38.71)	12 (48)	-	-	-	-	23 (74.19)	20 (80.00)	-	-	7 (22.58)	4 (16.00)	-	-	-	-	-	-	-	-	-	-	-	-	-	-	-	-	-	-
Ni et al. [30]	20 (39.22)	29 (60.42)	32 (62.75)	16 (33.33)			-	-	36 (70.59)	29 (60.42)	3 (5.88)	2 (4.17)	11 (25.57)	6 (12.50)	8 (15.69)	6 (12.50)	2 (3.92)	11 (22.92)	19 (37.26)	29 (60.42)	-	-	-	-	9 (17.65)	24 (50.00)	34 (66.67)	16 (33.34)	-	-	9 (17.65)	11 (22.92)
Odezimir et al. [31]	-	-	-	-					-	-	-	-	-	-	-	-	-	-	-	-	-	-	-	-	-	-	-	-	-	-	-	-
Ouyang et al. [32]	38 (88.37)	9 (75.00)	6 (13.95)	2 (16.67)			5 (11.63)	4 (33.34)	30 (69.77)	8 (66.67)	3 (6.98)	3 (25.00)	30 (69.77)	3 (25.00)	7 (16.28)	4 (33.34)	14 (32.56)	4 (33.34)	-	-	-	-	-	-	-	-	-	-	-	-	12 (27.91)	5 (41.67)
Pan et al. [33]	54 (57.45)	31 (67.39)	46 (48.94)	12 (26.09)	15 (15.96)	11 (23.91)	14 (14.89)	7 (15.22)	59 (62.77)	24 (52.17)	-	-	26 (27.66)	11 (23.91)	14 (14.89)	7 (15.22)	25 (26.60)	17 (36.96)	-	-	-	-	-	-	52 (55.32)	32 (69.57)	31 (32.98)	9 (19.57)	-	-	36 (38.30)	15 (34.61)
Pedraza et al. [34]	85 (59.03)	73 (48.35)	40 (27.78)	34 (22.52)	8 (5.56)	15 (9.93)	-	-	104 (72.23)	83 (54.97)	44 (30.56)	49 (32.45)	37 (25.70)	21 (13.91)	-	-	22 (15.28)	47 (31.13)	-	-	-	-	32 (22.23)	19 (12.58)	-	-	72 (50.00)	74 (49.00)	58 (40.28)	43 (28.48)	48 (33.34)	63 (41.72)
Pfaff et al. [35]	-	-	-	-	-	-	-	-	-	-	-	-	-	-	-	-	-	-	-	-	-	-	-	-	-	-	-	-	-	-	-	-
Shi et al. [36]	57 (46.72)	42 (39.62)	47 (47.00)	24 (30.38)	-	-	44 (44.00)	22 (27.85)	90 (88.24)	45 (56.96)	56 (56.57)	32 (40.51)	42 (42.00)	8 (10.13)	-	-	-	-	-	-	-	-	-	-	-	-	-	-	-	-	14 (11.48)	14 (13.21)
Singer et al. [37]	117 (54.93)	73 (49.33)	-	-	-	-	-	-	-	-	-	-	-	-	-	-	-	-	40 (18.78)	70 (47.30)	-	-	-	-	-	-	-	-	-	-	41 (19.25)	28 (18.92)
Su et al. [38]	56 (44.80)	63 (54.31)	54 (43.2)	37 (31.90)	-	-	31 (24.80)	18 (15.52)	92 (73.60)	61 (52.59)	17 (13.6)	18 (15.52)	34 (27.22)	17 (14.66)	-	-	-	-	52 (41.60)	77 (66.38)	-	-	-	-	60 (48.00)	70 (60.35)	61 (48.80)	40 (34.48)	-	-	33 (26.40)	44 (37.93)
Tajima et al. [39]	9 (39.13)	31 (67.39)	15 (65.22)	20 (43.48)	-	-	-	-	15 (65.22)	21 (45.65)	2 (8.70)	8 (17.39)	5 (21.74)	10 (21.74)	-	-	6 (26.09)	15 (32.61)	-	-	-	-	-	-	-	-	-	-	-	-	5 (21.74)	18 (39.13)
Tateishi et al. [40]	8 (40.00)	10 (66.67)	5 (25.00)	5 (33.34)	-	-	8 (40.00)	2 (13.33)	16 (80.00)	8 (53.34)	9 (45.00)	6 (40.00)	7 (35.00)	3 (20.00	5 (25.00)	2 (13.34)	3 (15.00)	2 (13.34)	-	-	-	-	-	-	-	-	-	-	-	-	9 (45.00)	11 (73.34)
Tonetti et al. [41]	26 (44.83)	17 (62.96)	23 (39.66)	7 (25.93)	1 (1.73)	0 (0.0)	9 (15.52)	18 (66.67)	47 (81.04)	20 (74.08)	35 (60.35)	12 (44.45)	19 (32.76)	4. (14.82)	11 (18.97)	2 (7.41)	13 (22.41)	6 (22.23)	-	-	-	-	-	-	-	-	-	-	-	-	16 (27.59)	17 (62.96)
Vatan et al. [42]	16 (51.61)	32 (48.49)	-	-	-	-	12 (38.71)	22 (33.34)	28 (90.32)	45 (68.18)	23 (74.19)	48 (72.73)	9 (29.03)	11 (16.67)	4 (12.90)	19 (28.79)	9 (29.03)	23 (34.85)	-	-	9 (29.03)	20 (30.30)	11 (35.48)	24 (36.36)	6 (19.36)	8 (12.12)	17 (54.84)	38 (57.58)	-	-	4 (12.90)	17 (25.76)
Wang et al. (1) [43]	18 (72.00)	21 (67.74)	4 (16.00)	5 (16.13)	13 (52.00)	15 (48.39)	-	-	20 (80.00)	23 (74.19)	4 (16.00)	3 (9.68)	13 (52.00)	16 (51.61)	4 (16.00)	4 (12.90)	13 (52.00)	17 (54.84)	10 (40.00)	6 (19.36)	-	-	-	-	-	-	-	-	-	-	-	-
Wang et al. (2) [44]	68 (61.26)	46 (59.74)	62 (55.86)	37 (48.05)	15 (13.51)	13 (16.88)	13 (11.71)	7 (9.09)	67 (60.36)	38 (49.35)	43 (38.74)	32 (41.56)	20 (18.02)	10 (12.99)	19 (17.12)	12 (15.59)	16 (14.42)	18 (23.38)	-	-	-	-	-	-	48 (43.24)	30 (38.96)	60 (54.06)	36 (46.75)	-	-	29 (26.13)	18 (23.38)
Xie et al. [45]	33 (80.49)	35 (77.78)	2 (4.88)	8 (1.78)	32 (78.05)	31 (68.89)	4 (24.00)	1 (15.76)	13 (31.71)	15 (33.34)	35 (85.37)	33 (73.34)	29 (70.73)	29 (64.45)	2 (4.88)	2 (4.45	25 (60.98)	24 (53.34)	-	-	-	-	-	-	29 (70.73)	30 (66.67)	5 (12.20)	9 (20.00)	-	-	16 (39.02)	14 (31.12)
Xu et al. [46]	114 (57.00)	136 (67.00)	59 (29.50)	43 (21.18)	-	-	48 (24.00)	32 (15.76)	112 (56.00)	93 (45.81)	-	-	36 (18.00)	25 (12.32)	29 (14.50)	20 (9.85)	55 (27.50)	70 (34.48)	-	-	43 (21.50)	36 (17.73)	-	-	114 (57.00)	127 (62.56)	47 (23.50)	49 (24.14)	99 (49.50)	73 (35.96)	46 (23.00)	34 (16.75)
Zang et al. [47]	24 (70.59)	18 (66.67)	12 (35.29)	6 (22.23)	-	-	3 (8.82)	3 (11.12)	18 (52.94)	10 (37.04)	6 (17.65)	5 (18.52)	7 (20.59)	4 (14.82)	-	-	13 (38.24)	8 (29.63)	-	-	-	-	-	-	13 (38.24)	11 (40.74)	16 (47.06)	9 (33.34)	-	-	14 (41.18)	11 (40.74)
Zhou et al. [48]	148 (53.43)	112 (54.51)	151 (54.51)	64 (34.41)	-	-	-	-	180 (64.98)	99 (53.23)	8 (2.89)	9 (4.83)	56 (20.22)	28 (15.05)	39 (14.08)	16 (8.60)	54 (19.50)	54 (29.03)	35 (12.64)	58 (31.18)	-	-	-	-	-	-	147 (53.07)	60 (32.36)	98 (35.38)	61 (32.80)	145 (52.35)	93 (50.00)

Abbreviations: n: number of patients; FR: futile recanalization; AF: atrial fibrillation; CVD: cardiovascular disease; HTN: hypertension; HL: hyperlipidemia; DM: diabetes mellitus; PS/TIA: prior stroke or transient ischemic attack; GC: good collaterals; APU: antiplatelet usage; ACU: anticoagulant usage; LAA: large-artery atherosclerosis; CE: cardioembolic; GA: general anesthesia; IVT: intravenous thrombolysis.

**Table 3 life-13-01965-t003:** Continuous Predictive Markers of Futile Recanalization Included in Meta-Analysis.

Author	Age		SBP		NIHSS		ASPECTS		OTT		OTR		Blood Glucose	
	FR (Mean ± SD)		FR (Mean ± SD)		FR (Mean ± SD)		FR (Mean ± SD)		FR (Mean ± SD)		FR (Mean ± SD)		FR (Mean ± SD)	
	Yes	No	Yes	No	Yes	No	Yes	No	Yes	No	Yes	No	Yes	No
Baek et al. [11]	-	-	-	-	-	-	-	-	-	-	-	-	-	-
Boisseou et al. [12]	72.6 (14.6)	66 (16.8)	151 (27.2)	143 (25.6)	18.7 (5.2)	14.0 (6.7)	6.3 (3.0)	7.7 (2.3)	-	-	305 (87.4)	288.7 (76.4)	7.2 (2.2)	6.5 (1.6)
Bouslama et al. [13]	64.9 (13.3)	61.8 (12.3)	153.6 (30.0)	158 (31.8)	-	-	-	-	673.2 (633.8)	598.3 (463.7)	-	-	-	-
Dhillon et al. [8]	-	-	-	-	18.1 (6.7)	15 (6.6)	-	-	403.8 (352.4)	338.5 (333.3)	-	-	-	-
Dong et al. [14]	-	-	-	-	-	-	-	-	-	-	-	-	-	-
Espinosa de Rueda et al. [15]	68.9 (11.4)	62.7 (13.3)	-	-	19.7 (4.5)	14 (6.1)	-	-	334.9 (152.9)	331 (252.5)	405.7 (170.5)	393.4 (259.1)	-	-
Feng et al. [16]	71.3 (10.0)	63.6 (11.1)	-	-	17.7 (5.3)	13 (4.5)	8.0 (1.5)	9 (1.5)	-	-	431.6 (174.6)	457.2 (195.9)	8.2 (3.3)	7.3 (2.4)
Gilberti et al. [17]	68.9 (13.3)	63 (14.5)	147.1 (19.3)	146.6 (20.1)	19.7 (3.2)	15 (6.1)	9.0 (1.6)	9 (1.5)	-	-	257.9 (72.3)	239.8 (71.2)	-	-
Hassan et al. [18]	70 (13.4)	65 (14.9)	-	-	18.2 (4.4)	15.9 (4.6)	8 (1.8)	8.6 (1.5)	283 (92.0)	228 (81.0)	336 (96.0)	273 (86.0)	-	-
Heitkamp et al. [19]	76.7 (13.4)	69 (14.9)	-	-	16.7 (5.2)	10 (5.2)	7.3 (2.2)	8.3 (2.2)	386.7 (267.3)	328.3 (232.1)	-	-	-	-
Hussein et al. [20]	59.3 (44.2)	60.3 (37.9)	149.5 (23.2)	144.9 (21.4)	20.3 (15.2)	16.3 (13.7)	5.7 (7.6)	6 (7.6)	265.8 (48.6)	239.2 (47.7)	-	-	7.6 (2.6)	6.7 (1.7)
Lattanzi et al. [21]	77.0 (10.5)	64.3 (17.4)	140.3 (19.6)	134.9 (20.9)	27 (11.0)	16 (10.0)	8.3 (2.3)	8.7 (2.3)	-	-	-	-	-	-
Lee et al. [22]	70.3 (12.1)	64 (11.8)	143.7 (27.3)	138.4 (27.0)	15.7 (5.2)	12.3 (6.7)	-	-	257.4 (110.9)	245.3 (124.0)	-	-	-	-
Liao et al. [23]	-	-	-	-	-	-	-	-	-	-	-	-	-	-
Lin et al. [24]	64.4 (13.1)	67.3 (10.8)	138.86 (25.08)	142.93 (23.87)	25.3 (14.6)	11.6 (10.2)	7.5 (2.3)	9 (1.5)	308.3 (140.9)	366.6 (235.5)	455.8 (217.2)	399.6 (170.9)	7.0 (2.2)	6.6 (1.7)
Linfante et al. [25]	-	-	-	-	-	-	-	-	-	-	-	-	-	-
Mechtouff et al. [26]	75 (13.0)	66 (15.0)	144.8 (24.4)	138.7 (20.5)	17 (6.1)	13.3 (7.5)	6.7 (2.3)	7.7 (1.5)	269 (153.0)	211.3 (118.6)	320 (195.2)	258.0 (128.6)	6.1 (1.2)	6.9 (1.7)
Meinel et al. [27]	76 (12.6)	69.0 (15.6)	152 (31.0)	149 (27.0)	18 (5.9)	13.7 (6.7)	7.7 (2.2)	8.3 (2.2)	243 (122.6)	232.3 (115.1)	-	-	7.4 (2.0)	6.6 (1.4)
Merlino et al. [28]	77.7 (9.0)	71.1 (11.1)	152.6 (24.6)	149.6 (21.6)	18.3 (3.7)	14 (6.0)	8.7 (3.0)	9 (2.3)	216.7 (75.0)	193.3 (63.9)	-	-	5.9 (0.6)	5.7 (0.5)
Mohammaden et al. [29]	68.4 (14.0)	65.2 (15.1)	-	-	21.3 (5.3)	13.8 (5.7)	9 (1.1)	9.4 (0.7)	-	-	-	-	-	-
Ni et al. [30]	76.7 (9.1)	66.3 (13.0)	152 (23.2)	144 (21.2)	18.7 (6.9)	12.7 (6.1)	7 (1.5)	8 (1.5)	603.7 (241.8)	564 (269.0)	-	-	6.9 (2.5)	6.3 (2.8)
Odezimir et al. [31]	-	-	-	-	-	-	-	-	-	-	-	-	-	-
Ouyang et al. [32]	69.0 (9.2)	64.7 (15.1)			30.0 (11.5)	30.5 (17.6)	8.0 (1.5)	9 (1.7)	563.3 (276.1)	366.7 (220.6)	678.3 (293.7)	518.7 (307.8)	8.8 (2.8)	8.1 (2.3)
Pan et al. [33]	71.0 (13.6)	63.7 (13.0)	152.7 (27.1)	144 (25.3)	16.0 (6.0)	11.3 (5.4)	8.7 (1.5)	8.7 (0.8)	-	-	-	-	7.9 (2.3)	7.8 (1.5)
Pedraza et al. [34]	74.4 (12.7)	68.4 (13.2)	-	-	18.1 (6.7)	15 (6.6)	8 (1.5)	9 (1.5)	-	-	424.3 (225.4)	201 (111.5)	-	-
Pfaff et al. [35]	-	-	-	-	-	-	-	-	-	-	-	-	-	-
Shi et al. [36]	69 (13.0)	62 (16.0)	150.3 (31.5)	138 (27.1)	19.3 (4.5)	17 (4.5)	-	-	264.7 (100.5)	233.7 (90.2)	349.3 (105.8)	327 (115.7)	-	-
Singer et al. [37]	71.0 (11.2)	61.7 (16.5)	-	-	-	-	-	-	-	-	-	-	-	-
Su et al. [38]	73.7 (10.5)	70.9 (10.2)	142.2 (24.3)	141.1 (26.4)	16.4 (4.7)	13 (3.0)	8.0 (1.5)	9 (1.5)	170.2 (64.1)	154.2 (48.4)	-	-	8.2 (3.1)	6.7 (1.6)
Tajima et al. [39]	78.5 (7.9)	72.6 (9.3)	-	-	23 (4.8)	19 (5.8)	6.8 (2.3)	7.9 (1.7)	-	-	302 (131.0)	244 (81.0)	-	-
Tateishi et al. [40]	72.3 (10.4)	62.7 (19.6)	-	-	18.7 (6.9)	12.7 (6.1)	7 (1.5)	8 (1.5)	603.7 (241.8)	579.7 (304.9)	73 (38.1)	73 (38.2)	-	-
Tonetti et al. [41]	-	-	-	-	-	-	-	-	-	-	-	-	-	-
Vatan et al. [42]	66 (13.0)	66 (12.0)	160 (35.0)	150 (30.3)	19 (3.1)	16.3 (3.0)	9.3 (0.8)	9.7 (0.8)	-	-	267.3 (94.8)	237.0 (103.8)	-	-
Wang et al. (1) [43]	66.4 (6.4)	64.8 (4.8)	-	-	-	-	-	-	-	-	454.4 (65.2)	416.0 (47.0)		
Wang et al. (2) [44]	71.0 (9.8)	64 (15.1)	146 (23.8)	140 (21.8)	16.7 (4.5)	12.0 (6.0)	-	-	-	-	420.0 (188.8)	400.0 (243.9)	7.6 (2.0)	6.7 (1.4)
Xie et al. [45]	65.3 (12.0)	62.3 (13.4)	-	-	27 (11)	16.0 (10)	-	-	-	-	859 (675.0)	545.0 (250.0)	9.8 (4.1)	9.8 (4.5)
Xu et al. [46]	68.4 (11.9)	60.1 (14.5)	-	-	17.7 (6.0)	15.3 (5.2)	7.7 (0.7)	7.7 (0.8)	273.0 (124.7)	268.7 (122.5)	-	-	7.0 (2.0)	6.4 (1.5)
Zang et al. [47]	63.3 (20.9)	47 (15.7)	131 (8.0)	125 (16.0)	17.3 (4.6)	12.7 (3.9)	7.3 (2.3)	8 (1.6)	257.4 (110.9)	245.3 (124.0)	349.3 (120.7)	372.0 (185.5)	7.5 (1.6)	6.8 (2.0)
Zhou et al. [48]	71.7 (11.2)	65.0 (12.0)	149 (23.8)	141.5 (26.1)	18.7 (6.0)	15.3 (5.2)	8 (3.0)	8.8 (1.9)	-	-	278.3 (73.0)	250.2 (73.2)	7.5 (2.2)	6.7 (1.7)

Abbreviations: FR: futile recanalization; n: number of patients; SBP: systolic blood pressure; NIHSS: National Institute of Health Stroke Severity; ASPECTS: Alberta Stroke Program Early CT Score; OTT: onset-to-treatment time; OTR: onset-to-recanalization time; BG: blood glucose; N: number of studies; n: number of patients.

**Table 4 life-13-01965-t004:** Meta-Analysis Results for Estimated Pooled Prevalence of Futile Recanalization: Summary Effects and Heterogeneity.

Subgroup	N	n	Crude Prevalence Rate	Pooled Prevalence Rate (from Meta-Analysis)	95% CI	z-Score	*p*-Value
Overall	39	11,700	49.28%	51%	0.48–0.54	47.66	*p* < 0.01
**Study design**							
Retrospective	22	5455	49.28%	51%	0.46–0.56	31.07	*p* < 0.01
Prospective	16	5883	48.70%	51%	0.47–0.56	33.46	*p* < 0.01
Prospective and retrospective	1	362	51%	-	-	-	-
**Region**							
Asia	17	3452	53.51%	53%	0.49–0.58	34.09	*p* < 0.01
Europe	10	4466	47.00%	48%	0.44–0.53	32.84	*p* < 0.01
North America	4	592	61.29%	62%	0.49–0.74	12.68	*p* < 0.01
Middle East	2	144	34.72%	35%	0.27–0.43	16.26	*p* < 0.01
Multiple	6	3049	46.21%	50%	0.41–0.59	13.71	*p* < 0.01

Abbreviations: N: number of studies; n: number of patients; CI: confidence interval.

**Table 5 life-13-01965-t005:** Meta-Analysis Results for Discrete Predictive Markers and Outcomes Associated with Futile Recanalization: Summary Effects and Heterogeneity.

Outcome	N	n	Effect Measure	Summary Effects	Heterogeneity ^¶^		Heterogeneity Variance Estimates			
				REDL						
				OR (95% CI)	Tests of Overall Effect	Cochran’s Q	H	I^2^≤ *	*p*-Value	τ^2^≤ ^Φ^
Male	34	9848	OR	0.87 [0.769; 0.973]	*p* = 0.016, z = −2.416	51.56	1.25	36% (95% CI: [0.0%; 60.1%])	*p* = 0.021	0.0355
AF	28	7471	OR	1.39 [1.223; 1.589]	*p* < 0.001, z = 4.976	36.01	1.155	25% (95% CI: [0.05%; 53.9%]	*p* = 0.115	0.0268
Alcohol	8	1104	OR	0.80 [0.581; 1.101]	*p* = 0.170, z = −1.372	3.93	0.75	0.0% (95% CI: [0.0%; 25.4%])	*p* = 0.787	0
CVD	17	2610	OR	1.15 [0.795; 1.671]	*p* = 0.454, z = 0.748	48.28	1.737	66.9% (95% CI: [14.6%; 82.5%])	*p* < 0.001	0.3634
HTN	33	9438	OR	1.65 [1.412; 1.924]	*p* < 0.001, z = 6.330	67.53	1.453	52.6% (95% CI: [0.5%; 72.4%])	*p* < 0.001	0.082
HL	28	6436	OR	0.97 [0.870; 1.088]	*p* = 0.627, z = −0.486	20.84	0.878	0.0% (95% CI: [0.0%; 21.1%])	*p* = 0.794	0
DM	33	9435	OR	1.71 [1.468; 1.990]	*p* < 0.001, z = 6.912	48.81	1.235	34.4% (95% CI: [0.0%; 58.9%])	*p* = 0.029	0.057
PS/TIA	16	4820	OR	1.30 [1.058; 1.592]	*p* = 0.012, z = 2.502	18.7	1.117	19.8% (95% CI: [0.0%; 57.3%])	*p* = 0.227	0.0287
Smoking	25	5595	OR	0.66 [0.572; 0.772]	*p* < 0.001, z = −5.349	29.01	1.099	17.3% (95% CI: [0.0%; 50.0%])	*p* = 0.220	0.0227
GC	7	1925	OR	0.33 [0.225; 0.486]	*p* < 0.001, z = −5.632	17.59	1.712	65.9% (95% CI: [0.0%; 86.4%])	*p* = 0.007	0.1659
APU	8	2935	OR	1.16 [0.976; 1.386]	*p* = 0.094, z = 1.676	4	0.756	0.0% (95% CI: [0.0%; 33.0%])	*p* = 0.779	0
ACU	8	2827	OR	1.33 [1.083; 1.634]	*p* = 0.007, z = 2.716	6.67	0.976	0.0% (95% CI: [0.0%; 54.1%])	*p* = 0.464	0
LAA	15	4083	OR	0.83 [0.671; 1.018]	*p* = 0.073, z = −1.793	22.02	1.264	36.5% (95% CI: [0.0; 66.7%])	*p* = 0.078	0.0549
CE	17	4841	OR	1.34 [1.100; 1.625]	*p* = 0.003, z = 3.016	31.36	1.4	49% (95% CI: [0.0%; 73.3%])	*p* = 0.012	0.0671
GA	7	5253	OR	1.53 [1.352; 1.737]	*p* < 0.001, z = 6.673	6.67	1.055	10.1% (95% CI: [0.0%; 62.8%])	*p* = 0.352	0.0031
IVT	30	9365	OR	0.75 [0.662; 0.857]	*p* < 0.001, z = −4.310	46.1	1.261	37.1% (95% CI: [0.0%; 62.0%])	*p* = 0.023	0.0381
sICH	17	4634	OR	7.37 [4.889; 11.116]	*p* < 0.001, z = 9.533	15.18	0.974	0.0% (95% CI: [0.0%; 41.6%])	*p* = 0.511	0
HT	11	2098	OR	2.98 [2.374; 3.746]	*p* < 0.001, z = 9.389	10.5	1.024	4.7% (95% CI: [0.0%; 53.5%])	*p* = 0.398	0.0073
Mortality	6	1018	OR	19.24 [1.573; 235.178]	*p* = 0.021, z = 2.315	22.9	2.14	78.2% (95% CI: [0.0%; 91.5%])	*p* < 0.001	7.6508

Abbreviations: AF: atrial fibrillation; CVD: cardiovascular disease; HTN: hypertension; HL: hyperlipidemia; DM: diabetes mellitus; PS/TIA: prior stroke or transient ischemic attack; GC: good collaterals; APU: antiplatelet usage; ACU: anticoagulant usage; LAA: large-artery atherosclerosis; CE: cardioembolic; GA: general anesthesia; IVT: intravenous thrombolysis; sICH: symptomatic intracranial hemorrhage; HT: hemorrhagic transformation; N: number of studies; n: number of patients; OR: odds ratio; CI: confidence interval; REDL: DerSimonian and Laird random-effects method; Q: heterogeneity measures were calculated from data with 95% confidence intervals (95% CI), based on noncentral X^2^ (common effect) distribution for Cochran’s Q test; H: relative excess in Cochran’s Q over its degrees of freedom; I^2^: proportion of total variation in effect estimate due to between study heterogeneity (based on Cochran’s Q test); τ^2^: between-study variance to test comparisons of heterogeneity among subgroups; *: values of I≤ are percentages; ^¶^: heterogeneity values were calculated from data with 95% CIs based on gamma (random-effects) distribution for Q; ^Φ^: heterogeneity variance estimates (τ^2^≤) were derived from the DerSimonian and Laird method.

**Table 6 life-13-01965-t006:** Meta-Analysis Results for Continuous Predictive Markers Associated with Futile Recanalization: Summary Effects and Heterogeneity.

Outcome	N	n	Effect Measure	Summary Effects	Heterogeneity ^¶^		Heterogeneity Variance Estimates			
				REDL						
				OR (95% CI)	Tests of Overall Effect	Cochran’s Q	H	I^2^≤ *	*p*-Value	τ^2 Φ^
Age	31	7417	SMD	0.49 [0.417; 0.564]	*p* < 0.0001, z = 13.033	59.67	1.41	49.7% (95% CI: [2.7%; 69.3%])	*p* = 0.001	0.0186
SBP	18	4841	SMD	0.20 [0.127; 0.266]	*p* < 0.001, z = 5.538	20.86	1.11	18.5% (95% CI: 0.0%; 54.5%)	*p* = 0.233	0.0039
NIHSS	29	8892	SMD	0.75 [0.648; 0.857]	*p* < 0.001, z = 14.088	124.87	2.122	77.6% (95% CI: [42.9%, 88.1%])	*p* < 0.001	0.0535
ASPECTS	22	5588	SMD	−0.37 [−0.464; −0.271]	*p* < 0.001, z = −7.471	52.01	1.574	59.6% (95% CI: [3.4%; 78.0%])	*p* < 0.001	0.0265
OTT	18	6964	SMD	0.22 [0.131; 0.304]	*p* < 0.001, z = 4.925	38.5	1.505	55.8% (95% CI: [0.0%; 77.7%])	*p* = 0.002	0.0153
OTR	17	2716	SMD	0.38 [0.185; 0.565]	*p* < 0.001, z = 3.863	83.57	2.285	80.9% (95% CI: [46.8%; 90.2%])	*p* < 0.001	0.1182
BG	17	5099	SMD	0.31 [0.217; 0.409]	*p* < 0.001, z = 6.367	35.91	1.498	55.4% (95% CI: [0.0%; 77.1%])	*p* = 0.003	0.0191

Abbreviations: SBP: systolic blood pressure; NIHSS: National Institute of Health Stroke Severity; ASPECTS: Alberta Stroke Program Early CT Score; OTT: onset-to-treatment time; OTR: onset-to-recanalization time; BG: blood glucose; N: number of studies; n: number of patients; OR: odds ratio; CI: confidence interval; REDL: DerSimonian and Laird random-effects method; Q: heterogeneity measures were calculated from data with 95% confidence intervals (95% CI), based on noncentral X^2^ (common effect) distribution for Cochran’s Q test; H: relative excess in Cochran’s Q over its degrees of freedom; I^2^: proportion of total variation in effect estimate due to between study heterogeneity (based on Cochran’s Q test); τ^2^: between-study variance to test comparisons of heterogeneity among subgroups; *: values of I≤ are percentages; ^¶^: heterogeneity values were calculated from data with 95% CIs based on gamma (random-effects) distribution for Q; ^Φ^: heterogeneity variance estimates (τ^2^≤) were derived from the DerSimonian and Laird method.

## Data Availability

The original contributions presented in the study are included in the article and online Supplemental Information, and further inquiries can be directed to the corresponding author.

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
