# Peer review of "Comprehensive Meta-Analysis of Futile Recanalization in Acute Ischemic Stroke Patients Undergoing Endovascular Thrombectomy: Prevalence, Factors, and Clinical Outcomes"

_life, 2023, doi:10.3390/life13101965_

Round 1

Reviewer 1 Report

The review is important for the medical field  and to understand Comprehensive Meta-Analysis of Futile Recanalization in 2 Acute Ischemic Stroke Patients Undergoing Endovascular 3 Thrombectomy.  However, a careful review of the minor English language is necessary because there  misplaced paragraphs.

Figure and tables are so complex, please simplify if possible.

Simplify the whole manuscript that you have written.

Specific references must be added which are comply with the topic.

Describe the significance of statistical analysis.

some more recent references must be added in introduction.

Careful review of the minor English language is necessary because there  misplaced paragraphs.

Author Response

We thank the reviewer for the careful review of our work and insightful comments. Based on the remarks, we have made signficant changes to the manuscript overall to improve readability and structure. 

  1. Whilst, the content of Tables and Figures, we believe, are important; we have now formatted it so as to improve it's presentation.
  2. The whole manuscript has been simplified.
  3. The abstract has been revised.
  4. The Introduction has been improved and more latest references of interst have been included and cited, as appropriate.
  5. We have highlighed the need for this analyses. 

We would like to thank again the reviewer for the time and feedback. Please find the revised manuscript attached. 

Reviewer 2 Report

I really thank the editor for this opportunity to revise such an interesting paper. The cohort of patients is huge and from a clinician point of view the importance of have in mind factors determining or avoiding a futile revascularization in acute ischemic stroke patient is very interesting, as well as the pooled rate of 51% that is important to have in mind. 

Nevertheless some changing, not in methods but mainly in the quality of reporting data and more discussion should be inserted in the paper, since the quality of what is reported is very high. 

Abstract

Well realized not many concerns. 

Line 40: I would refer that the pooled rate of 51% is about the number of FR. 

In this section I would not report all the abbreviation in the brackets for the clinical factors. 

Line 56-57: onset-to-recanalization and onset-to-treatment time is the increasing of the time? So the longer it takes, the worse is the result? In this section is not clear to the reder. 

Conclusions: The substantial prevalence of FR around 51% …. Otherwise does not understand how much is in conclusion. 

Methods: 

Paragraph 2.1 and 2.2 repeat some concepts and therefore might be fused in the same paragraph. (e.g. studies not in English etc)

Results: 

Line 224: I would just erase the reference to the crude percentage of 49.3% since is not useful in a proportion meta-analysis and will procedure just confusion. 

Just report the pooled rate. 

Line 291: I would just take out the comment! (it is a results section!) to the crude percentage.

Same in line 308 and in line 317. 

Same, I would just erase line 324, 325

Paragraph 3.15-9 are redundant and copy and paste of the same format. Just make an unique paragraph and remind to the Figure 2. 

At the same time, paragraph 3.1.- 3.4 are really difficult to read and report data already in Tables and figures. 

I would just write paragraph on “predictive indicators for FR” “predictive indicators avoiding FR”. 

Same for the clinical outcomes following FR: very interesting aspect, please just make one unique paragraph. 

Discussion:

Line 700: erase the crude percentage. What is the necessity?

As a general consideration you should report here, apart the fact that is the larger number MA on FR, what is good and what is important and different in this MA on Fr and what is the take home message. 

You should discuss reason on how some risk factors enhance FR and on the other hand how other seems to limit this occurrence. 

Lines from 735 to 757 are just a list of things that are missing and not adequately reported in the paper? Just hard to read and follow, even if the content is of great value. 

Conclusions
Please remove abbreviation in the conclusion of risk factors. Just make some general comments or reports the most relevant. 

Quality of english is enough. The results section MUST be varied otherwise it is just a copy and paste of results from Tables and the same format really annoying to the reader. 

The discussion section must be renovated since it is pretty much a list of items and does not convey enough details and insights of the paper. 

Author Response

We thank the reviewer for the extensive review of our work and positive feedback and comments. Based on the suggestions, we have made best of efforts to improve the manuscript. The key changes are highlighted below;

  1. The pooled rate has been included as suggested
  2. The reference to crude rates have been removed.
  3. The abstract has been revised, and sentences have been restructued.
  4. The Introduction has been improved and more latest references of interest have been included and cited, as appropriate.
  5. The results section has been concised - especially the predictors and clinical outcomes sub-sections.
  6. The Discussion has also been updated to discuss implications of our findings and redundant sections have been removed.
  7. The abbreviations have been removed from the Conclusion, as suggested.

We would like to thank again the reviewer for the time and feedback. Please find the revised manuscript attached.

Reviewer 3 Report

Authors presented a very interesting Meta-Analysis of Futile Recanalization in Acute Ischemic Stroke Patients Undergoing Endovascular Thrombectomy. The paper aims to comprehensively assess the pooled prevalence of Futile Recanalization, considering several factors related to it, and establish the association of FR with long-term clinical outcomes among acute ischemic stroke patients undergoing to endovascular thrombectomy.

The meta-analysis is complete, the PRISMA checklist and flowchart are present, the objective is clear and the findings are concise and interesting. 

Author Response

We thank the reviewer for the time and positive review of our work and insightful comments. We have revised the manuscript to further improve the readability. Please find the revised manuscript attached.
